# Amplitude mode in the planar triangular antiferromagnet $Na_{0.9}MnO_2$

Rebecca L. Dally[1,2], Yang Zhao [3,4], Zhijun Xu[3,4], Robin Chisnell[3], M.B. Stone [5], Jeffrey W. Lynn[3], Leon Balents[6] & Stephen D. Wilson[1]

Amplitude modes arising from symmetry breaking in materials are of broad interest in condensed matter physics. These modes reflect an oscillation in the amplitude of a complex order parameter, yet are typically unstable and decay into oscillations of the order parameter's phase. This renders stable amplitude modes rare, and exotic effects in quantum antiferromagnets have historically provided a realm for their detection. Here we report an alternate route to realizing amplitude modes in magnetic materials by demonstrating that an antiferromagnet on a two-dimensional anisotropic triangular lattice ($\alpha$-$Na_{0.9}MnO_2$) exhibits a long-lived, coherent oscillation of its staggered magnetization field. Our results show that geometric frustration of Heisenberg spins with uniaxial single-ion anisotropy can renormalize the interactions of a dense two-dimensional network of moments into largely decoupled, one-dimensional chains that manifest a longitudinally polarized-bound state. This bound state is driven by the Ising-like anisotropy inherent to the $Mn^{3+}$ ions of this compound.

[1] Materials Department, University of California, Santa Barbara, CA 93106, USA. [2] Department of Physics, Boston College, Chestnut Hill, MA 02467, USA. [3] NIST Center for Neutron Research, National Institute of Standards and Technology, Gaithersburg, MD 20899, USA. [4] Department of Materials Science and Engineering, University of Maryland, College Park, MD 20742, USA. [5] Neutron Scattering Division, Oak Ridge National Laboratory, Oak Ridge, TN 37831, USA. [6] Kavli Institute for Theoretical Physics, University of California, Santa Barbara, Santa Barbara, CA 93106, USA. Correspondence and requests for materials should be addressed to S.D.W. (email: stephendwilson@ucsb.edu)

Many of the seminal observations of amplitude modes in magnetic materials arise from quantum effects in one-dimensional antiferromagnetic chain systems when interchain coupling drives the formation of long-range magnetic order[1]. For instance, bound states observed in the ordered phases of $S = 1$ Haldane systems[2–4] or in the spinon continua of $S = 1/2$ quantum spin chains[5–9] were shown to be longitudinally polarized and reflective of the crossover into an ordered spin state. While the chemical connectivity of magnetic ions in these systems is inherently one-dimensional, alternative geometries such as planar, anisotropic triangular lattices can also in principle stabilize predominantly one-dimensional interactions in antiferromagnets[10]. In the simplest case, geometric frustration in a lattice comprised of isosceles triangles promotes dominant magnetic exchange along the short leg of the triangle while the remaining two equivalent legs frustrate antiferromagnetic coupling between the chains. The result is a closely spaced two-dimensional network of magnetic moments whose dimensionality of interaction is reduced to be quasi one-dimensional.

A promising example of such an anisotropic triangular lattice structure is realized in α-phase NaMnO₂. Layered sheets of edge-sharing $MnO_6$ octahedra are separated by layers of Na ions, and the orbital degeneracy of the octahedrally coordinated $Mn^{3+}$ cations ($3d^4$, $t_{2g}^3 e_g^1$ valence) is lifted via a large, coherent Jahn–Teller distortion[11]. This distorts the triangular lattice such that the leg along the in-plane $b$-axis is contracted 10% relative to the remaining two legs. As a result, the $S = 2$ spins of the $Mn^{3+}$ ions decorate a dimensionally frustrated lattice where one-dimensional intrachain coupling along $b$ is favored and interchain coupling is highly frustrated. This spin lattice eventually freezes into a long-range ordered state below $T_N = 45$ K[11]; however, previous studies of powder samples have suggested an inherently one-dimensional character to the underlying spin dynamics[12]. Such a scenario suggests an intriguing material platform for the stabilization of an amplitude mode in a conventional spin system (i.e., one with diminished local moment fluctuations and a quenched Haldane state[13]) as the ordered state is approached, and a static, staggered mean field is established.

In this paper, we present single crystal neutron scattering data that show that the planar antiferromagnet α-Na₀.₉MnO₂ exhibits quasi one-dimensional spin fluctuations that persist into the AF ordered state. Additionally, our data reveal that an anomalous, dispersive spin mode appears as AF order sets in, and that this new mode is longitudinally polarized with an inherent lifetime limited by the resolution of the measurement. This longitudinally polarized-bound state demonstrates the emergence of a magnetic amplitude mode in a spin system where geometric frustration lowers the dimensionality of magnetic interactions and amplifies fluctuation effects. Intriguingly, this occurs in a compound where strong quantum fluctuations inherent to $S = 1/2$ systems and singlet formation effects inherent to integer-spin Haldane systems —both typical settings for longitudinal-bound state formation— are absent. To explain the stabilization of this amplitude mode, we present a model that captures the excitation as a two-magnon-bound state whose binding energy derives from an easy-axis single-ion anisotropy inherent to the orbitally quenched $Mn^{3+}$ ions. This anisotropy orients the moments along a preferred axis and renders them Ising-like. Our work establishes α-Na$_x$MnO₂ and related lattice geometries as platforms for realizing unconventional spin dynamics in a dense network of one-dimensional antiferromagnetic spin chains[14,15].

## Results

**Crystal and spin structures of α-Na₀.₉MnO₂.** To demonstrate the emergence of an amplitude mode in α-Na₀.₉MnO₂, careful

descriptions of the lattice and spin structures are first necessary. We note here that units for wave vectors throughout the manuscript are given in reciprocal lattice units $(H, K, L)$ where $\mathbf{Q}\left[\text{Å}^{-1}\right] = \left(\frac{2\pi}{a \sin \beta} H, \frac{2\pi}{b} K, \frac{2\pi}{c \sin \beta} L\right)$ and $a$, $b$, $c$, and $\beta$ are the lattice parameters of the unit cell. Figure 1a shows the projection of the

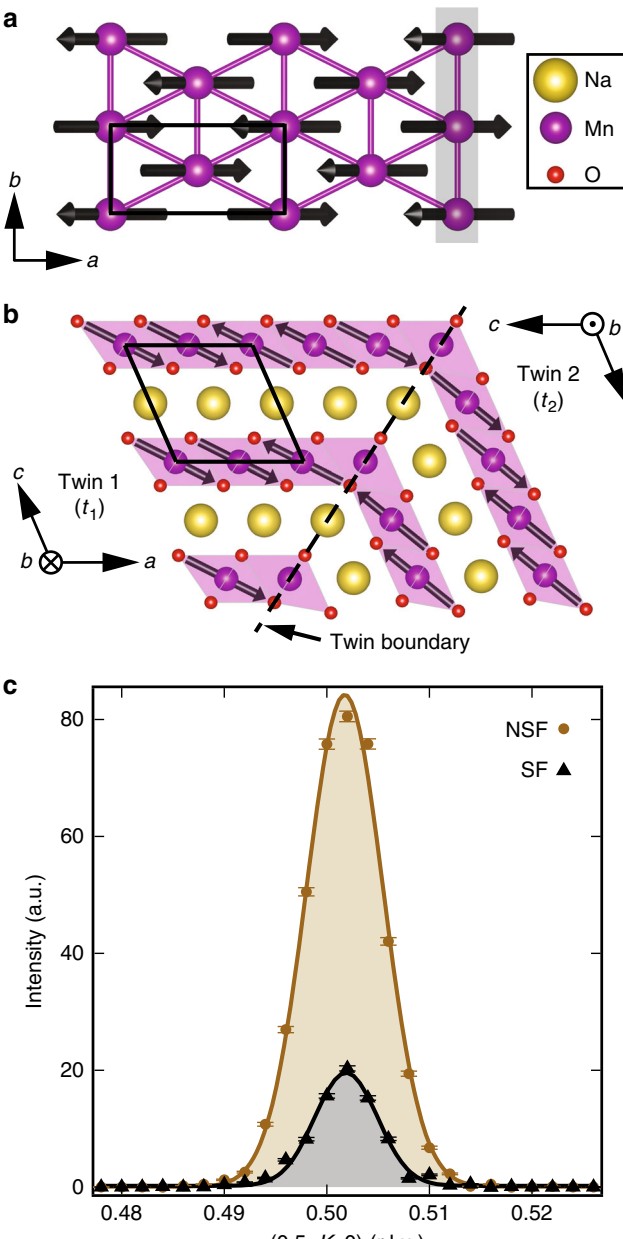

**Fig. 1** Summary of the crystal and magnetic structures of α-NaMnO₂. **a** Projection of the 3D magnetically ordered state onto the $ab$-plane. The black rectangle denotes the chemical unit cell. The gray shaded region highlights the chain direction between nearest neighbor Mn atoms. **b** Moments in the $ac$-plane, as well as the intergrowth of two twin domains. The black rhombus again denotes the chemical unit cell. Polarized elastic neutron data at the antiferromagnetic zone center in **c** confirm the orientation of the magnetic moments within the ordered state ($T = 2.5$ K) as previously reported. The shaded regions are the Gaussian fits of the non spin-flip (NSF) and spin-flip (SF) channels, which were used to determine the moment orientation. The neutron polarization $\mathbf{P}$ is perpendicular to the scattering vector and parallel to the $c$-axis in this configuration. Error bars represent one standard deviation

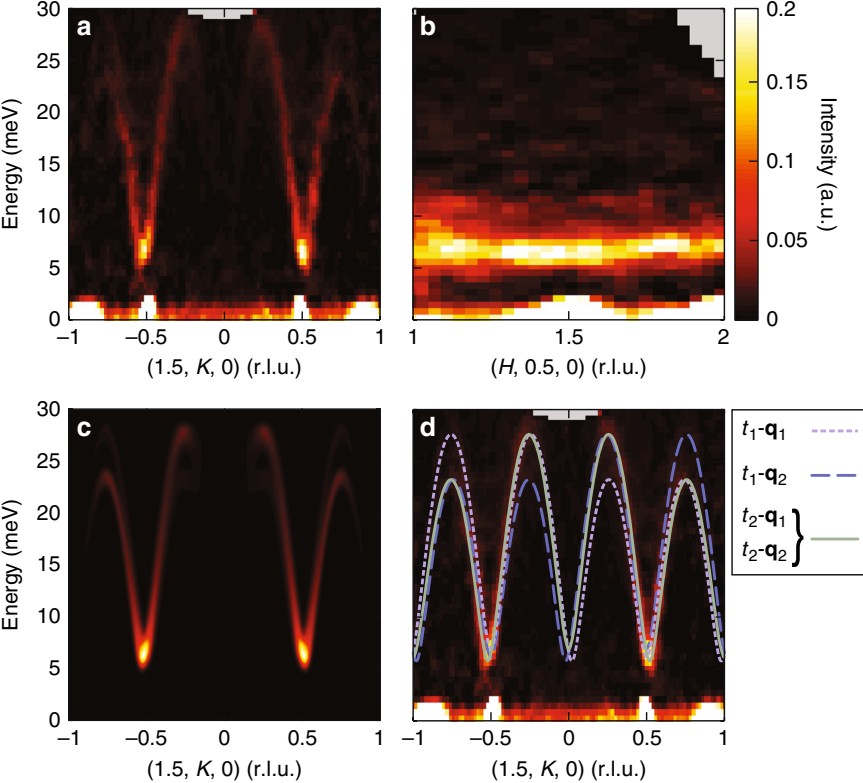

**Fig. 2** Magnon spectra at $T = 2.5$ K collected via time-of-flight neutron scattering measurements. **a** Dispersion of magnons along the chain axis ($K$-direction) across the full bandwidth of excitations. Data were integrated across $-0.1$ to $0.1$ along $L$ and from $1.4$ to $1.6$ along $H$. **b** Dispersion along the interchain axis ($H$-direction) with data integrated from $-0.1$ to $0.1$ along $L$ and from $0.48$ to $0.52$ along $K$. **c** Simulated scattering intensities using the model fit to the data described in the main text and integrated over the same values as **a**. **d** Fit transverse modes from the four allowed domains generating the spectral weight in **c** and then overplotted with the raw data from **a**

low temperature, ordered spin lattice of α-NaMnO$_2$ onto the *ab*-plane where one spin domain with propagation vector $\mathbf{q}_1 = (0.5, 0.5, 0)$ is illustrated. The antiferromagnetic chain direction is shaded in gray. Due to the degeneracy of the frustrated interchain coupling, a second spin domain with propagation vector $\mathbf{q}_2 = (-0.5, 0.5, 0)$ also stabilizes, and the moments in both domains are oriented approximately along the $[-1, 0, 1]$ apical oxygen bond direction due to an inherent uniaxial single-ion anisotropy[11]. The large Jahn–Teller distortion renders the lattice structure prone to crystallographic twinning[16] and the relative orientations of the moments in the two resulting crystallographic twins, twin 1 ($t_1$) and twin 2 ($t_2$) are depicted in Fig. 1b. This results in four allowed domains: $t_1 - \mathbf{q}_1$, $t_1 - \mathbf{q}_2$, $t_2 - \mathbf{q}_1$, and $t_2 - \mathbf{q}_2$. Crucially, the moments in both crystallographic and magnetic twin domains are nearly parallel to a common $[-1, 0, 1]$ axis. This is verified via polarized elastic neutron scattering measurements in the AF state at $T = 2.5$ K shown in Fig. 1c. These data demonstrate that in a single domain, $t_1 - \mathbf{q}_1$, probed at $\mathbf{Q} = (0.5, 0.5, 0)$ the moments are rotated ~7° away from the $[-1, 0, 1]$ axis within the *ac*-plane. We note here that previous studies of stoichiometric NaMnO$_2$ have reported an extremely subtle distortion into a lower triclinic symmetry below the antiferromagnetic transition[11]. Our neutron diffraction measurements fail to detect this distortion in the average structure of Na$_{0.9}$MnO$_2$ crystals, and we therefore analyze data using the higher symmetry monoclinic structure. A recent report suggests that the triclinic phase occurs only as an inhomogeneous local distortion[17]; hence our inability to observe the reported triclinic distortion may arise from its absence in the average structure, its suppression due to the Na vacancies in our samples, or due to the resolution threshold of our measurements. Despite this ambiguity, the reported triclinic

distortion in NaMnO$_2$ is subtle and would generate roughly a 0.12% difference between next-nearest neighbor (interchain) exchange pathways, which can be neglected for the purposes of the present study.

**Spin Hamiltonian of α-Na$_{0.9}$MnO$_2$.** In order to understand the interactions underlying the AF ground state of this system, inelastic neutron scattering measurements were performed. Spin excitations measured within the ordered state about the AF zone centers, $\mathbf{Q} = (1.5, \pm 0.5, 0)$, are shown in Fig. 2. Inspection of the momentum distribution of the spectral weight reveals that the magnetic fluctuations underpinning the AF state at $T = 2.5$ K are quasi one-dimensional. Figure 2a demonstrates that the magnetic excitations along the in-plane $K$-axis, parallel to the short leg of the triangular lattice, show an anisotropy gap at the zone center and a well-defined dispersion; however, the magnon dispersion in directions orthogonal to this axis are diffuse. Specifically, the spin waves dispersing between the MnO$_6$ planes (along $L$) are dispersionless (see Supplementary Fig. 2) as expected for the planar structure of α-Na$_{0.9}$MnO$_2$, and Fig. 2b shows that spin wave energies dispersing perpendicular to the AF chain direction in the plane (along $H$) are only weakly momentum dependent. This demonstrates that the spin fluctuations exist as quasi-one-dimensional planes of scattering in $(\mathbf{Q}, E)$ space, driven by the strong interchain frustration inherent to the lattice and consistent with the large magnetic frustration parameter of this compound[18].

As twin effects from both crystallographic and spin domains may obscure any subtle dispersion along $H$ due to interchain interactions, inspection of zone boundary energies and analysis of the full bandwidth are necessary to quantify the weak interchain

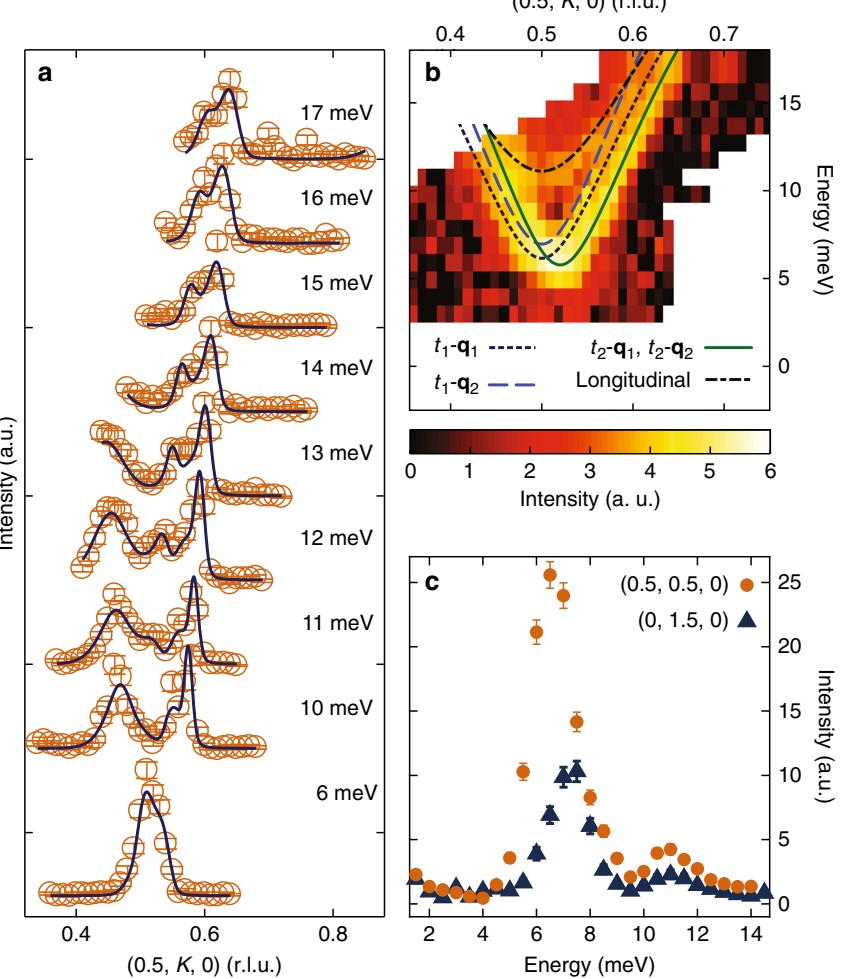

**Fig. 3** Inelastic neutron scattering data at $T = 2$ K revealing an additional zone center mode. **a** Momentum scans at various energies through $\mathbf{Q} = (0.5, 0.5, 0)$. Solid lines are resolution convolved fits to the data as described in the text. **b** An intensity color map summarizing the scattering data from **a** along with dispersion of modes comprising the fits to the data in **a**. Lines show fits to the expected transverse modes from the different crystallographic and magnetic domains within the sample: $t_1$-$\mathbf{q}_1$ (dashed purple), $t_1$-$\mathbf{q}_2$ (dashed blue), and $t_2$-$\mathbf{q}_1$ and $t_2$-$\mathbf{q}_2$ (solid green). The black dashed line represents the dispersion fit to the longitudinal mode in the spectrum as described in the text. **c** Constant energy scans at the three dimensional (0.5, 0.5, 0) and quasi one-dimensional (0, 0.5, 0) AF zone centers. Error bars in **a** and **c** represent one standard deviation

exchange terms. Therefore, in order to parameterize the dispersion measured in Fig. 2a, the high energy data were analyzed using a four-domain model ($t_1 - \mathbf{q}_1$, $t_1 - \mathbf{q}_2$, $t_2 - \mathbf{q}_1$, $t_2 - \mathbf{q}_2$) as well as by fitting lower energy triple-axis data shown in Fig. 3 and Supplementary Fig. 3. The data were modeled using the single-mode approximation and the spin Hamiltonian, $H = J_1 \sum_{nn} \mathbf{S}_i \cdot \mathbf{S}_j + J_2 \sum_{nnn} \mathbf{S}_i \cdot \mathbf{S}_j - D \sum_n \left(S_n^z\right)^2$, where $J_1$ is the twofold nearest neighbor exchange coupling, $J_2$ is the fourfold next-nearest neighbor coupling, and $D$ is a uniaxial, Ising-like, single-ion anisotropy term. The dispersion relation generated from a linear spin wave analysis of this Hamiltonian is given by $E(\mathbf{Q}) = S\sqrt{\omega_\mathbf{Q}^2 - \lambda_\mathbf{Q}^2}$, where $\omega_\mathbf{Q} = 2(J_1 + D + J_2\cos(\pi H + \pi K))$ and $\lambda_\mathbf{Q} = 2(J_1 \cos(2\pi K) + J_2\cos(\pi H - \pi K))$ (see Supplementary Note 1 and Supplementary Fig. 1 for details), and the results from fitting the data yielded a $J_1 = 6.16 \pm 0.01$ meV, $J_2 = 0.77 \pm 0.01$ meV, and $D = 0.215 \pm 0.001$ meV. These values are roughly consistent with earlier powder averaged measurements of spin dynamics in $\alpha$-NaMnO$_2$, although these earlier measurements were not sensitive enough to resolve a $J_2$ term[12].

The magnon modes from the four-domain model are over-plotted as lines with the raw time-of-flight data in Fig. 2d, and the

total simulated intensities summed from all modes are shown in Fig. 2c. Good agreement is seen between the data in Fig. 2a and the simulated intensities shown in Fig. 2c and Supplementary Fig. 4. To further illustrate this model at lower energies closer to the zone center gap value, data collected via a thermal triple-axis spectrometer are shown in Fig. 3. Momentum scans through the AF zone center are plotted in Fig. 3a at energies from $\Delta E = 3$ meV to 18 meV with the resulting color map of intensities plotted in Fig. 3b. Magnon modes dispersing from the four domains in the system using the same $J_1$–$J_2$–$D$ model described earlier and convolved with the instrument resolution function are plotted as solid lines fit to the data in Fig. 3a. Crucially, unlike the model presented in Fig. 2, describing this lower energy data also requires the introduction of one additional dispersive mode. This mode is distinct from the transversely polarized magnons anticipated in this material, and it represents an unexpected longitudinally polarized-bound state as described in the next section.

**Longitudinally polarized mode.** To more clearly illustrate the appearance of an additional mode in the low energy spin dynamics, a magnetic zone center energy scan at $\mathbf{Q} = (0.5, 0.5, 0)$ is plotted in Fig. 3c. This scan shows the large buildup of spectral

   

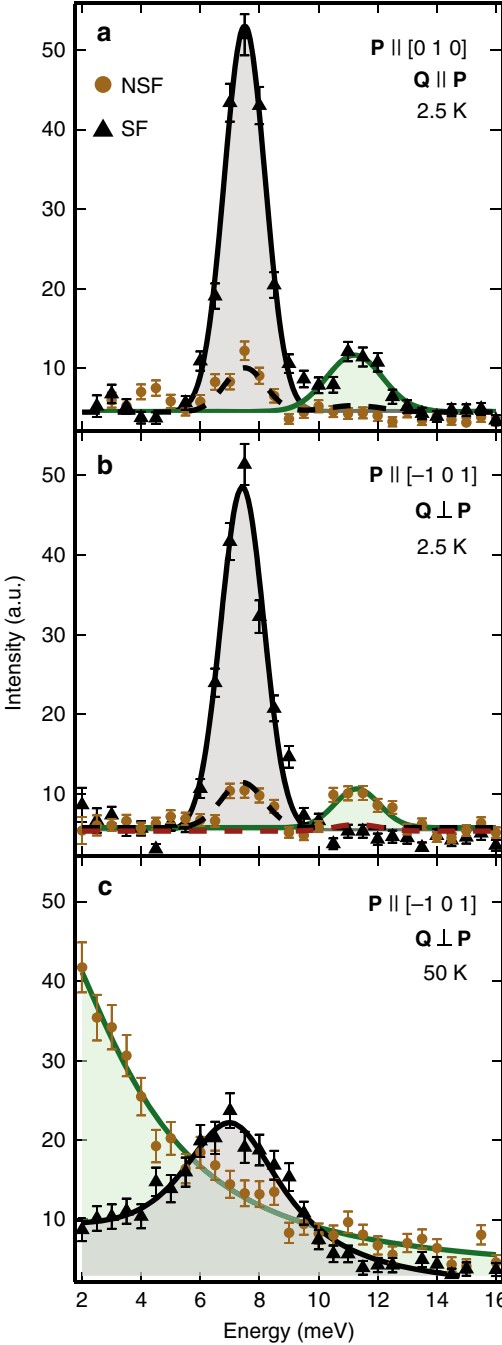

**Fig. 4** Polarized inelastic neutron scattering data about the quasi-1D zone center $\mathbf{Q} = (0, 1.5, 0)$. **a** Data collected with the neutron polarization $\mathbf{P}$ parallel to $\mathbf{Q}$. The two modes appear only in the spin-flip (SF) channel and so are both magnetic in origin. **b** Data collected with the neutron polarization $\mathbf{P}$ parallel to the $[-1, 0, 1]$ axis. Transverse spin fluctuations appear in the SF channel, and longitudinal fluctuations appear in the non spin-flip (NSF) channel. Data in both **a** and **b** were taken in the 3D ordered state at $T = 2.5$ K, and black dashed lines indicate the expected bleed through from the SF channel into the NSF channel due to imperfect neutron polarization. The red dashed line in **b** represents the expected bleed through from the NSF channel into the SF channel. **c** The same configuration as **b** but above the antiferromagnetic transition temperature at $T = 50$ K. Solid lines in **a** and **b** are Gaussian fits parameterizing each mode. The solid line for the NSF channel in **c** is a Lorentzian fit centered at $\Delta E = 0$ meV and the solid line for the SF channel in **c** consists of two Lorentzian fits (one centered at $\Delta E = 0$ meV and one at the single magnon gap energy). Error bars represent one standard deviation

weight above the $\Delta E = 6.15 \pm 0.04$ meV zone center gap, consistent with the quasi one-dimensional magnon density of states, and above this gap a second zone center mode near 11 meV appears. This 11 meV mode is not accounted for by any of the expected transverse modes in this system, and it is also quasi one-dimensional in nature. Figure 3c demonstrates negligible inter-chain dispersion as $\mathbf{Q}$ is rotated from the three dimensional $\mathbf{Q} = (0.5, 0.5, 0)$ to one-dimensional $\mathbf{Q} = (0, 0.5, 0)$ AF zone center, and the 11 meV mode's dispersion along the chain direction is plotted in Fig. 3b. While there is a limited bandwidth ($\Delta E = 11$–15 meV) where this new mode remains resolvable inside of the dispersing transverse magnon branches, the narrow region of dispersion was empirically parameterized using a one-dimensional $J$–$D$ model with $H = J \sum_n \mathbf{S}_n \cdot \mathbf{S}_{n+1} - D \sum_n (S_n^z)^2$ and $E(\mathbf{Q}) = \sqrt{\Delta^2 + c^2 \sin^2(2\pi K)}$. The gap value from this para-meterization was fit to be $\Delta = 11.11 \pm 0.06$ meV and $c = 21.6 \pm 0.1$ meV. The dispersion fit to this higher energy mode along with the dispersions fit to the transverse magnon modes are over-plotted with the data in Fig. 3b. We again note that this additional 11 meV dispersive mode was incorporated within the fits shown in Fig. 3a.

To further explore the origin of the anomalous 11 meV branch of excitations near the AF zone center, polarized-neutron scattering measurements were performed using an experimental geometry that leveraged the quasi one-dimensional nature of the spin excitations. Specifically, magnetic excitations were measured about the one-dimensional zone center, $\mathbf{Q} = (0, 1.5, 0)$. As the magnetic moment $\mathbf{\mu}$ is oriented nearly parallel to the $[-1, 0, 1]$ crystallographic axis, two transversely polarized magnon modes along the $[1, 0, 1]$ and the $[0, 1, 0]$ directions are expected in the ordered state, each carrying an oscillation of the orientation/phase of the staggered magnetization. The $(\mathbf{Q} \times \mathbf{\mu} \times \mathbf{Q})$ orientation factor in the neutron scattering cross section renders it only sensitive to the components of the moments' fluctuations perpendicular to $\mathbf{Q}$, and thus transverse spin waves observed at the $\mathbf{Q} = (0, 1.5, 0)$ are dominated by $[1, 0, 1]$ polarized modes. By further orienting the neutron's spin polarization, $\mathbf{P}$, parallel to $\mathbf{Q}$, all allowed magnetic scattering is guaranteed to appear in the channel where the neutron's spin is flipped during the scattering process[19]. Fig. 4a shows the results of energy scans collected at $\mathbf{Q} = (0, 1.5, 0)$ with data collected in both the spin-flip (SF) and non spin-flip (NSF) channels. As expected, peaks from the $\Delta E = 6.15$ and $\Delta E = 11.11$ meV zone center modes appear only in the SF cross-sections (dashed lines indicate the transmission expected by the polarization efficiency of SF scattering into the NSF channel and vice versa).

Using the same scattering geometry but with the neutron polarization now rotated parallel to the $[-1, 0, 1]$ direction, the magnetic scattering processes polarized along the $[1, 0, 1]$ axis (i.e., the resolvable transverse spin wave mode) should remain in the SF channel while scattering processes polarized parallel to the neutron polarization direction (i.e., nearly parallel to the ordered moment direction) will instead appear in the NSF channel. Figure 4b shows the results of energy scans with $\mathbf{P}||[-1, 0, 1]$ where the 11 meV mode now appears only in the NSF channel and the 6 meV mode remains only in the SF channel. Again, the small amount of intensity around 6 meV in the NSF channel can be explained by the calculated contamination of scattering from the SF channel into the NSF channel. This demonstrates that the 11 meV mode and the associated upper branch of spin excitations are polarized longitudinally, reflecting an amplitude mode of the staggered magnetization, while the 6 meV mode and the lower energy branch of excitations are polarized transverse to the moment direction. Keeping $\mathbf{P}||[-1, 0, 1]$, an identical energy scan collected at $T = 50$ K in the paramagnetic state shows that the

   

coherent amplitude mode vanishes for $T > T_{AF}$ and critical fluctuations driving the phase transition dominate the longitudinal spin response (Fig. 4c). Conversely, the transverse modes remain well-defined at high temperatures, reflective of the strong, inherently one-dimensional coupling and the single-ion anisotropy of the Mn moments.

## Discussion

Earlier neutron measurements have demonstrated that the ordered moment of α-NaMnO$_2$ (2.92 $\mu_B$) is significantly reduced from the classical expectation (4 $\mu_B$)[11], suggesting substantial fluctuation effects in this material. Additionally, ESR experiments find evidence of strong low temperature fluctuations in the ordered state[18]. The $\Delta S = S - \langle S_z \rangle = 0.54$ missing in the static ordered moment, however, can be accounted for when integrating the total inelastic spectral weight in earlier powder inelastic neutron measurements[12]. Neutron scattering sum rules therefore imply that the ratio of the momentum and energy integrated weights of the longitudinal and transverse spin fluctuations should be $\frac{\Delta S(\Delta S+1)}{(S-\Delta S)(2\Delta S+1)} = 0.27$[20], which is greater that the ratio of the intensities of the zone center modes $I_{long}/I_{tran} = 0.19$. This rough comparison suggests that, while the amplitude mode observed in our measurements is relatively intense, it remains within the bounds of the allowed spectral weight for longitudinal fluctuations.

Relative to stoichiometric α-NaMnO$_2$ powder samples, Na vacancies in the α-Na$_{0.9}$MnO$_2$ crystal studied here are unlikely to generate this long-lived, dispersive amplitude mode. Each Na vacancy naively binds to a hole on the Mn-planes and creates an Mn$^{4+}$ $S = 3/2$ magnetic impurity. The corresponding hole is bound to the impurity site due to strong polaron trapping. As this state remains localized within the lattice[21,22], the role of vacancies can be viewed as introducing random static magnetic impurities within the MnO$_6$ planes. Neither this random, static disorder nor the high density of twin boundaries inherent to the lattice[23] are capable of directly generating a coherent spin mode; however, they may indirectly contribute to destabilizing the ordered Néel state, pushing α-Na$_{0.9}$MnO$_2$ closer to a disordered regime and enhancing fluctuation effects.

For an easy-axis antiferromagnetic chain at 0 K, two degenerate, gapped transverse magnon modes are expected as Néel order sets in; however the amplitude mode observed in the ordered state of α-Na$_{0.9}$MnO$_2$ is unexpected. Since this longitudinally polarized mode has a zone center lifetime that is constrained by the resolution of the spectrometer ($\Delta E_{res} = 2.25$ meV at $E = 11$ meV) without an observable high energy tail and with an energy below twice the transverse modes' gap values, it likely arises as a long-lived two-magnon-bound state[24,25]. The finite binding energy of this state as determined by $E_{bind} = 2E_{gap} - E_{long} = 1.2 \pm 0.1$ meV at the zone center $\mathbf{Q} = (0.5, 0.5, 0)$ implies an attractive potential between magnons once Néel order is established.

To explore this further, we theoretically consider the existence of a bound state within a one-dimensional model, neglecting $J_2$. We further consider only zero temperature for simplicity, and perform a semi-classical large $S$ analysis based on anharmonically coupled spin waves around an antiferromagnetically ordered state. For $J_1 e^{-S} \ll D \ll J_1$, and $S \gg 1$, the one-dimensional chain is ordered at $T = 0$, which allows $J_2$ to be neglected without qualitative errors. As detailed in Supplementary Note 2, the resulting description applies: (1) the anisotropy induces a single magnon gap $\Delta = 2S\sqrt{2J_1 D}$ so that the magnon dispersion near the magnetic zone center is that of a relativistic massive particle with $E = \sqrt{v^2 k^2 + \Delta^2}$ with magnon velocity $v = 2J_1 S$, and (2) the dominant interaction between opposite spin magnons with

momentum $k \ll \sqrt{\frac{D}{J_1}}$ is an attractive delta-function of strength $U = -2J_1$. This problem has a bound state which is in the non-relativistic limit described by two one-dimensional particles of mass $M = \frac{\Delta}{v^2}$, for which the textbook result for the binding energy is $E_{bind} = \frac{MU^2}{4}$. Using the values above, we obtain $\frac{E_{bind}}{\Delta} = \frac{1}{4S^2}$. This is the leading result for large $S$, and in the limit $D/J_1 \ll 1$. While this limit predicts a binding energy approximately 3 times smaller than that observed for α-Na$_{0.9}$MnO$_2$, moving away from this limit and incorporating the non-negligible $D$ in the system can account for this discrepancy. We note that, while we performed calculations in the one-dimensional model for simplicity, this is only a matter of convenience rather than essential physics: the magnons and their dispersion and interactions evolve smoothly upon including $J_2$, which would be necessary to model the spectrum at $T > 0$.

The long-lived amplitude mode in the Néel state of α-Na$_{0.9}$MnO$_2$ is distinct from those observed in canonical 1D integer-spin chain systems such as CsNiCl$_3$[4], where the longitudinal mode emerges as the Haldane triplet state splits due to an internal staggered mean field. The Haldane state within the frustration-driven $S = 2$ spin chains in α-Na$_{0.9}$MnO$_2$ is easily quenched under small anisotropy[13,26,27], and α-Na$_{0.9}$MnO$_2$ is thought to be outside of the Haldane regime. Calculations predict that the phase boundary between the $S = 2$ Haldane state and antiferromagnetic order appears at $D/J = 0.0046$ (for easy-axis $D$)[27], far away from the experimentally measured $D/J = 0.035$ in Na$_{0.9}$MnO$_2$. While amplitude modes in other quantum spin systems close to singlet instabilities have also been recently reported in the quasi-two-dimensional spin ladder compound C$_9$H$_{18}$N$_2$CuBr$_4$[28] and the two-dimensional ruthenate Ca$_2$RuO$_4$[29], the formation of a longitudinal-bound state in α-Na$_{0.9}$MnO$_2$ is distinct from modes in these and other $S = 1/2$ spin chain systems possessing substantial zero-point fluctuations[8].

Instead, in α-Na$_{0.9}$MnO$_2$, the interplay of geometric frustration and the Jahn–Teller quenching of orbital degeneracy uniquely conspire to create a quasi-one-dimensional magnon spectrum that condenses due to the attractive potential provided by an Ising-like single-ion anisotropy. As a result, α-Na$_{0.9}$MnO$_2$ provides an intriguing route to realizing an intense, stable amplitude mode in a planar AF. The dimensionality reduction realized within its chemically two-dimensional lattice also suggests that other α-NaFeO$_2$ type transition metal oxides[30,31] possessing coherent Jahn–Teller distortions may host similarly stable amplitude modes, depending on their inherent anisotropies. More broadly this class of materials presents an exciting platform for exploring unconventional-bound states such as bound soliton modes[14] stabilized in a quasi-one-dimensional spin setting.

## Methods

**Crystal growth and characterization**. Na$_2$CO$_3$ and MnCO$_3$ powders (1:1 ratio plus 10% weight excess of Na$_2$CO$_3$) were mixed and sintered in an alumina crucible at 350 °C for 15 h, reground and sintered for an additional 15 h at 750 °C. Dense polycrystalline rods were made by pressing the powder at 50,000 psi in an isostatic press. The rod was sintered in a vertical furnace at 1000 °C for 15 h and then quenched in air, before being transferred to a four mirror optical floating zone furnace outfitted with 500 W halogen lamps. The crystals were grown at a rate of 20 mm h$^{-1}$ in a 4:1 Ar:O$_2$ environment under 0.15 MPa of pressure. Inductively coupled plasma atomic emission spectroscopy (ICP-AES) was used to determine the Na/Mn ratio and to check that the expected mass of Mn was present. The ratio of Mn$^{3+}$/Mn$^{4+}$ was determined through X-ray absorption near edge spectroscopy (XANES), X-ray photoelectron spectroscopy (XPS), and $^{23}$Na solid-state NMR (ssNMR). Detailed crystal growth and characterization can be found in Dally et al.[23] Samples were handled as air-sensitive and stored in an inert environment. Time outside of an inert environment was minimized (e.g., during crystal alignment for neutron scattering experiments). The crystal faces are flat, and no degradation of the surface was observed during alignment. The same ~0.5 g crystal was used for both neutron TOF and triple-axis experiments.

**Time-of-flight experimental setup**. Neutron time-of-flight data in Fig. 2 and Supplementary Figs. 2b and 4 were taken at the Spallation Neutron Source at Oak Ridge National Laboratory using the instrument SEQUOIA. The sample was sealed in a He-gas environment, mounted in a cryostat, and aligned in the $(H, K, 0)$ horizontal scattering plane. All data were taken at 4 K with an incident energy $E_i = 60$ meV, and the fine-resolution fermi chopper rotating at 420 Hz. For data collection, the sample was rotated through a range of 180° with 1° steps. A background scan was collected by removing the sample from the neutron beam and collecting the scattering from the empty can.

**Time-of-flight data analysis**. An aluminum only (empty can) background was subtracted from all data before plotting. SpinW[32], a Matlab library, was used to simulate the magnetic excitations for the TOF data. Given the spin Hamiltonian, magnetic structure and twinning mechanisms (structural and magnetic), SpinW uses linear spin wave theory to numerically calculate and display the dispersion. The simulation was convolved with the energy resolution function of the neutron spectrometer ($E_i = 60$ meV, $F_{chopper} = 420$ Hz). Simulations for Fig. 2c were run over the same range that the data were binned for Fig. 2a and d (i.e., $1.4 < H < 1.6$ and $-0.1 < L < 0.1$), and then averaged together.

**Triple-axis experimental setup**. Triple-axis neutron data were taken with the instrument, BT7[33,34], at NCNR with a PG(002) vertically focused monochromator and the horizontally flat focus mode of the PG analyzer system. PG filters before and after the sample were used during collection of elastic data ($E_i = 14.7$ meV), and only a PG filter after the sample was used during inelastic operation (fixed $E_f = 14.7$ meV). Unpolarized data from Fig. 3 and Supplementary Fig. 3 were taken with open$-25'-50'-120'$ collimations (denoting the collimation before the monochromator, sample, analyzer, and detector, respectively), and the sample was aligned in the $(H, K, 0)$ scattering plane. Supplementary Fig. 2a data were unpolarized and taken with open$-50'-50'-120'$ collimators in the $(H, H, L)$ plane. Polarized data in Fig. 1c were taken with open$-25'-25'-120'$ collimation in the $(H, K, 0)$ scattering plane. Polarized data in Fig. 4 were taken in the $(H, K, H)$ plane with open$-80'-80'-120'$ collimations.

**Triple-axis polarization efficiency corrections**. It was determined that only two (one NSF and one SF) of the available four neutron scattering cross-sections were needed for polarization analysis. Flipping ratios were taken throughout the experiment at the $(2, 0, 0)$ nuclear Bragg peak and at all temperatures probed. These flipping ratios were used to correct for the polarization efficiency.

**Triple-axis data analysis**. All data were normalized to the neutron monitor counts, $M$. Error bars represent one standard deviation of the data. For unpolarized data, this was calculated by the square root of the number of counts, $\sqrt{N}$, where $N$ is the number of counts. The lower monitor counts in polarized data were considered by propagating the error in the monitor counts, $\sqrt{M}$, such that $\sigma^2 = \frac{N}{M^2}\left(1 + \frac{N}{M}\right)$. The determination of the moment angle utilized the polarized elastic data shown in Fig. 1c. After correcting for the polarization efficiency, the integrated intensities of the NSF and SF peaks were found by fitting the data to Gaussian functions. These intensities were used to find the moment angle following the technique in Moon et al.[19]

Fits to the constant energy scans (unpolarized inelastic neutron scattering data) in Fig. 2d, 3a, b and Supplementary Fig. 3 used the Cooper–Nathans approximation[35] in ResLib[36], a program that calculates the convolution of the spectrometer resolution function with a user supplied cross section. The cross section used for the transverse excitation was the single-mode approximation of a two-dimensional spin lattice with single-ion anisotropy, as described in Supplementary Note 1. Cross-sections for $t_1 - \mathbf{q}_1$, $t_1 - \mathbf{q}_2$, $t_2 - \mathbf{q}_1$, and $t_2 - \mathbf{q}_2$ were all included during the fitting routine using the relation between the first moment sum rule and the dynamical structure factor,

$$\int_{-\infty}^{\infty} (\hbar\omega) S^{\alpha\alpha}(\mathbf{q}, \hbar\omega) \mathrm{d}(\hbar\omega) = -\sum_{n,\beta} J_n [1 - \cos(\mathbf{q} \cdot \mathbf{a}_n)]\left(1 - \delta_{\alpha\beta}\right)\langle\langle S^{\beta}_{\mathbf{R}_j}, S^{\beta}_{\mathbf{R}_j + \mathbf{a}_n}\rangle\rangle.$$

The contribution to the scaling factor from the single-ion term is small[37], and therefore, was not included. The longitudinal excitation was empirically fit using the single-mode approximation for a one-dimensional chain, given its unresolvable dispersion along $H$. The longitudinal mode gap was determined by fitting the data in the range where it was resolvable (below 18 meV). This gap value was fixed and the fitting routine was run again, allowing all other parameters to vary. A single intrinsic HWHM for all excitations was refined during fitting and refined to be negligibly small. Additionally, a scaling prefactor was also refined for each crystallographic twin, where the two magnetic domains within a crystallographic twin were assumed to have the same weight (i.e., $t_1 - \mathbf{q}_1$ and $t_1 - \mathbf{q}_2$ had the same prefactor).

Polarized inelastic neutron data in Fig. 4 are plotted as raw data, uncorrected for polarization efficiency. The dashed lines representing the expected bleed through from the SF channel into the NSF channel in Fig. 4a and b were determined from the measured flipping ratio.

**Data availability**. The data that support the findings of this study are available from the corresponding author upon reasonable request.

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

## Acknowledgements

S.D.W. and R.L.D. acknowledge assistance in characterizing samples from Raphaële Clément. S.D.W. and R.L.D. gratefully acknowledge support from DOE, Office of Science, Basic Energy Sciences under Award DE-SC0017752. Work by L.B. was supported by the DOE, Office of Science, Basic Energy Sciences under Award No. DE-FG02-08ER 46524.

## Author contributions

R.L.D. synthesized the $\alpha$-NaMnO$_2$ single crystals and analyzed the data. R.L.D., Y.Z., Z. X., R.C., M.B.S., and J.W.L. helped perform neutron experiments. L.B. performed theoretical analysis of the spin dynamics. R.L.D. and S.D.W. designed the neutron experiments. R.L.D., L.B., and S.D.W. prepared and wrote the manuscript.

## Additional information

**Competing interests:** The authors declare no competing interests.

