## [Peer Review File · Nature Communications]

Reviewers' comments:

Reviewer #1 (Remarks to the Author):

In this article, Dally et al. experimentally study the magnetic properties of $\text{Na}_{0.9}\text{MnO}_2$ in the alpha-phase using neutron scattering. At sufficiently low temperature, the magnetic moments order. However, it is known that the dynamics of the system is mostly one-dimensional due to the geometric frustration of the lattice.

After having confirmed the magnetic order orientation at low temperature, the authors measure the excitation spectrum, uncovering both transverse and longitudinal excitations. Both excitations are gapped (the gap of the longitudinal mode being about twice as big as that of the transverse mode), and the dispersion relation of the transverse mode is well explained by a 1D Heisenberg model with single ion anisotropy (in agreement with previous studies). The authors then give further evidence that the higher energy branch indeed corresponds to longitudinal excitations by analyzing the effect of the neutrons polarization.

Although the experimental results are interesting, I find their analysis quite weak. Indeed, after the thorough description of the experimental data, we are left with a very short, and somewhat vague, discussion of the nature of the longitudinal mode. For example, no attempt is made to compare the excitation spectrum of the longitudinal mode with that of a simple spin-wave theory (for instance, following an approach similar to that of Ref. 24). Would this kind of approach allow for an explanation the longitudinal gap? And if not, is there a good reason?

The discussion of the Ginzburg-Landau theory is also not satisfying. Following Ref. 25, the authors discuss the lifetime and mass of the longitudinal mode in terms of the proximity to a quantum critical point (QCP). Yet the authors remark in their introduction that their compound is far from any known QCP, which renders the discussion hard to follow. I would like to remark that being far from a QCP does not imply a short-lived longitudinal mode (it is rather that one can show that in some models, being close to a QCP insures a well-defined amplitude mode). Indeed, the longitudinal mode could be long-lived because it cannot kinematically decay into two transverse modes (which, as far as I understand, could be the case here, since twice the gap of the transverse mode is larger than that of the longitudinal mode).

Therefore, I think that the term "anomalous" to describe the longitudinal mode should be substantiated much more. As it is, the article is too fact-focused, and I cannot recommend it for publication, unless the authors provide an improved analysis of their data.

I also have some additional questions/remarks:

1) I do not understand why the longitudinal and transverse energy scans are so different above T_N (Fig. 4c). Since $D \ll T_N$, and since there isn't any magnetic order, I would naively expect the fluctuations to be much more isotropic. Could the authors comment on that?

2) Units for wave-vectors should be given.

3) The vector P in Fig. 1 is defined only later in the article.

Reviewer #2 (Remarks to the Author):

This manuscript presents a study of the excitations in α - $\text{Na}_0.9\text{MnO}_2$. The study is motivated by two reasons: the recent success on the growth of a single crystal of this compound and two recent publications/preprint reporting so-called amplitude modes in two-dimensional (2D) systems [Tao arXiv:1705.06172v1, Jain, A. Nature Physics 13, 633–637 (2017)]. The nature of the excitations in this compound is interesting in the field of condensed matter physics, as it can give important information to further understand the phase diagram of a one-dimensional (1D) spin chain with significant single ion anisotropy. The authors claim that the nature of these excitations can have a bigger impact on broader areas, as the findings reported by H. Tao and A. Jain do. The authors present in this manuscript experimental results of the excitations of this compound, based on polarized neutron scattering measurements. Two gapped excitations are measured, both of them with a 1D nature. The lowest energy mode is a transverse mode, which can be understood and described using linear spin wave theory. The study of this excitation –in a powder sample- was already published by Stock, et al PRL 103, 077202 (2009). The higher energy excitation is a longitudinal mode –amplitude mode-, which has not been reported before, and its nature is unknown. The authors do not present any clear explanation on the existence of this mode and the different scenarios are shortly listed (in page 14) although they have not been explored. Which brings me to think that the authors do not understand the physics behind this compound. I find a couple of irregularities in this manuscript and I consider that the data analysis has not been done meticulously and cannot be used to claim any conclusion, apart from just reporting the existence of the longitudinal mode. Therefore, I consider that this manuscript, in its current form, should not be published in Nature communications. My specific comments are as follow:

- The title is misleading as it mentions a 2D triangular antiferromagnet, but the material is an effective spin chain (1D). This is explained in the abstract where the authors claim that a new route to realize an amplitude mode in a 2D spin arrangement is to have an effective 1D magnet. This statement falls far from any logic. Please be aware that when talking about dimensionality in magnetism, we refer to the effective magnetic exchange interactions and those do not necessarily coincide with the atomic distribution in the lattice.
- The abstract does not represent the results. It mentions “Higgs bosons” without them being relevant in this compound as it is far away from a quantum critical point. It discusses 2D magnets, although this system is quasi-1D. It does not describe all their observations and does not provide a conclusion.
- The sample reported here is not stoichiometric. However, previous published studies were done in stoichiometric powder samples. It is confusing the mixed of sample names throughout the manuscript as the stoichiometry is only discussed at the end of the manuscript. The authors should make a comment about this much earlier in the text.
- Diffuse magnetic scattering is mentioned in page 7. The existence of this signal was also previously mentioned by Stock et al. However, it is not possible, or it is not clear in the figures where this intensity is.
- The dark blue color in colorplots (Fig. 2a,c) is misleading. Please make this white as I guess there is no available data in those regions.
- Fig 2 d: The 6meV mode has an unusual sudden drop in the intensity which is not symmetric along Q. The authors make no comment regarding this.
- Have the authors looked for higher energy modes or there is only data below 13meV at (0.5,0.5,0)?
- The authors compare the data to two structure factors in Fig. 3. They make their first conclusion on the nature of the higher energy mode, saying that the longitudinal function describes better the data, that making it different to a transversely polarized spin wave excitation. However, both structure factors fail completely to describe the data. There is also a clear modulation of the intensity as a function of Q –damped sine- but the authors fail on recognizing it.

- The authors use the model –linear spin wave theory- already published by Stock in order to describe the dispersion relation of the transverse mode. They shortly mention in page 8 that the higher energy mode was also fit with the same J , but do not give details on the dispersion relation for this mode. Do they also use linear spin wave theory to describe this mode? They give no comment on which model they use, on how it works and how can they use a model without understanding the physics of the compound.
- In page 15 they quickly mentioned that the material is thought to be far from the Haldane regime. However, looking at DMRG results which calculate the phase diagram for $s=2$ Haldane chains with single ion anisotropy, the reported values of $D=0.196\text{meV}$ and $J=5.8\text{meV}$ do not look too far from the Haldane regime (Phys. Rev. B 87, 235106, 2013)

Reviewer #3 (Remarks to the Author):

The authors present detailed unpolarised and polarised inelastic neutron scattering results for the $S = 2$ triangular-lattice compound $\alpha\text{-NaMnO}_2$. They find a gapped spin-wave (transverse) mode and an additional excitation with predominantly longitudinal polarisation and a gap approximately double that of the transverse branch. They discuss the context of the amplitude (longitudinal) mode in quantum and “not-so-quantum” antiferromagnets.

This referee has no grounds for complaint concerning the experimental analysis, which appears to be solid and delivers a convincing demonstration of the existence of a longitudinal (amplitude) mode. However, the theoretical discussion is unclear and in places self-contradictory.

From as early as the title, the authors refer to their material as a classical spin system. Aside from the fact that all spins are at minimum semiclassical (their origin in electron spin and their description by ladder (raising and lowering) operators), this is definitely not the case for $S = 2$. Here one may consider the clear presence of a Haldane gap in the $S = 2$ AF chain, to which the manuscript refers. If the authors wish to argue for the effects of the D term in reducing quantum fluctuations, this is certainly not clear in the early part of the discussion. Later the authors even report “substantial fluctuation effects” in the ordered moment, which reduce it significantly from its classical value.

On language, although the authors call the amplitude mode in magnetic materials a “condensed matter analogue” of the Higgs boson, this is only half an analogy because the amplitude mode lacks any gauge character (in this sense an amplitude mode of a superconducting order parameter is closer). Further, the primary new aspect of the present work is to extend the study of amplitude modes beyond the highly spin-symmetric cases discussed to date, but such an easy-axis system would seem to have only discrete symmetries, and thus the idea of a mode emerging from the breaking of a continuous symmetry is also lost.

In several places the authors refer to their system as being “far from a quantum critical point,” also referring to a “quantum disordered regime.” First, these two concepts have to be an oxymoron in a classical spin system (first point). It is correct that an ordering transition is sufficient for the appearance of an amplitude mode, and thus they are possible in a classical system (thermal transition). However, if the authors wish to be guided by previous work on the amplitude mode near a QCP, such as Ref. [25], one of the things they will need is new language. Second, it is not at all clear that this system need be far from a 1D to 2D transition: because it has only discrete and not continuous spin symmetries, the nature of the putative QCP need not be similar to the situation in Ref. [25] and it is not a given that the mode masses would vanish at the transition. Thus the gap alone cannot be used as an indicator of proximity to a QCP.

The manuscript refers both at the beginning and end to topological excitations and (bound) solitons. What do the authors have in mind here and do they really have anything to say about

either topic ? At minimum, some remarks of justification are required. If the former refers to Haldane-type string order, how does this fit in with the spin being "classical" ? On the latter, again if the authors are motivated by the discussion of domain walls (also kinks or solitons) in Ising systems, several steps of logic are missing from the manuscript.

On P4 there are some curious formulations: 1) it seems to be stated that the amplitude mode is detected "as ordered state is approached," but in fact it is only present when a magnetic order parameter is established. 2) the authors refer to a "longitudinal bound state," but this seems to presuppose one of the possible scenarios discussed later. Is it clear, particularly in the present low-symmetry system, that the longitudinal mode has to be considered a bound mode rather than as something "elementary" ? 3) there is the first occurrence of the statement about being "far from any known QCP," which as noted above contains two unjustified assumptions. 4) there are two unqualified statements about this being a "classical spin system."

On smaller matters:

As a matter of curiosity, what is the bandwidth of the modes shown in Fig. 2a ? What happens in the upper half of the Brillouin zone ? The question of mixing with two-boson excitations is an important and potentially informative one.

The Neel temperature of 45 K suggests an effective interchain interaction (in 3D) on the order of 4 meV. Can the authors explain why nothing like this is visible in Fig. 2b ? Zero effective interchain interaction due to perfect frustration should also be reflected in T_N .

The non-stoichiometry of the system would appear to present a significant source (10%) of disorder. The discussion of this point seems a little backwards – given that Na is a very generous electron donor, it is curious to state that its absence creates hole, and in fact thinking in a language of holes (mobile objects in semiconductors) is curious in an insulating magnet. The possible impact of 10% disorder is dismissed very quickly and this point would benefit from a deeper discussion; this referee does agree that there is no argument for the disorder to create a dispersive spin mode, and in fact that the authors could be using it to strengthen their arguments for semiclassical behaviour [a) the system behaves as though there is little difference between the $S = 2$ and $S = 3/2$ sites; b) if the 10% reduction is included then the expected ordered moment is already rather less than $4\mu_B$ (P13), meaning that the fluctuation effect is not as strong as it appears]. However, there remains a serious non-sequitur where the authors refer to the regime of no magnetic order as "quantum disordered," having claimed that the spin is classical.

On P14 the authors suggest that the amplitude mode could be lighter than the gapped transverse modes. Do they have a mechanism in mind for this ? If the effective binding energy of 2 transverse modes were so strong that the amplitude mode is lighter than them, should one not review theoretical picture of which excitations qualify as "elementary" and which as "compound" ?

In the very last sentence the authors also speculate about NaMO₂, with no references. Do they have some solid justification in mind for this ? How much is known about spin anisotropies and materials non-stoichiometries in any other such system ? It seems that any change of the $S = 2$ spin will create something significantly more or less quantum than the current compound, with effects that cannot be foreseen too clearly from one known case.

In summary, the manuscript reports solid experimental work but the discussion of its context could use considerable improvement before it could be deemed suitable for publication in something like its current form.

Reviewer #4 (Remarks to the Author):

The authors present evidence of a 1D longitudinally polarized magnon bound state on single crystal of the 2D antiferromagnetic material NaMnO₂. Motivated by the report of 1D spin dynamics inferred by measurement on powder samples (ref 11), the authors employ inelastic neutron scattering experiments on single crystal to unveil a dispersive mode along the b^* direction at nearly twice the energy of the lowest energy branch.

Further measurements using polarized neutron beams using different polarization direction with respect to momentum direction at the $(0,3/2,0)$ reciprocal space position demonstrates that this mode is longitudinally polarized and has a long lifetime, limited by the instrumental resolution. This result is new, and the use of polarization analysis provides a direct experimental evidence for the polarization direction of the mode, and strongly supports the claim.

The paper is well written, concise, and certainly of interest for the magnetism community. I feel at this stage however that the publication would immensely benefit from some theoretical backup that could reproduce at least qualitatively the energy and dispersion of the mode in the (Q,w) space probed by the experiment and explain how this unusual mode differs from what founds in other antiferromagnets. Alternatively, the authors need to explain why it is not possible at present to compute the dispersion and intensity of this mode with available methods or what are the current theoretical limitations. For example in the $S=1/2$ quantum system CuSO₄·5D₂O a very precise computation of the high-order spinon states had been provided and agrees remarkably well with the data (Nature Physics 9, 435–441 (2013)). Without this additional information, I am not in a position to support publication in Nature Communication at this stage.

Dear Editor,

Thank you for forwarding the referee reports for our manuscript. We have now substantially revised the manuscript including new measurements and added theory as well as an updated discussion in order to address the referee's comments and questions regarding our results. The changes to the manuscript are detailed in our replies to the referees' comments below. Please note also that there is now an accompanying supplemental material section to the paper, modified figures, two additional authors, and a modified title (as requested by referee). We feel the substantially strengthened manuscript is now suitable for publication in Nature Communications. Thank you very much for your consideration.

Reviewer #1 (Remarks to the Author):

In this article, Dally et al. experimentally study the magnetic properties of Na_{0.9}MnO₂ in the alpha-phase using neutron scattering. At sufficiently low temperature, the magnetic moments order. However, it is known that the dynamics of the system is mostly one-dimensional due to the geometric frustration of the lattice. After having confirmed the magnetic order orientation at low temperature, the authors measure the excitation spectrum, uncovering both transverse and longitudinal excitations. Both excitations are gapped (the gap of the longitudinal mode being about twice as big as that of the transverse mode), and the dispersion relation of the transverse mode is well explained by a 1D Heisenberg model with single ion anisotropy (in agreement with previous studies). The authors then give further evidence that the higher energy branch indeed corresponds to longitudinal excitations by analyzing the effect of the neutrons polarization.

Although the experimental results are interesting, I find their analysis quite weak. Indeed, after the thorough description of the experimental data, we are left with a very short, and somewhat vague, discussion of the nature of the longitudinal mode. For example, no attempt is made to compare the excitation spectrum of the longitudinal mode with that of a simple spin-wave theory (for instance, following an approach similar to that of Ref. 24). Would this kind of approach allow for an explanation the longitudinal gap? And if not, is there a good reason?

We thank the reviewer for his/her careful review of our manuscript and for noting the novelty of our experimental results. The manuscript is now substantially revised both with more data characterizing the spin Hamiltonian and with new theory that models the amplitude mode as a two-magnon bound state.

To address the referee's specific suggestion of comparing our results with simple spin wave theory (such as in Ref. 24), a comparison between our results and the expectations of trivial two-magnon scattering at the AF zone center is plotted below. The zone center gap for the primary domain (the lower energy of the two spin domains) was chosen and we simulated the two-magnon dispersion using a one dimensional model with easy-axis anisotropy. This model has the maximum divergence in the magnon DOS and is a good limiting case to test our data against. For illustration purposes, the J and D values fit

to the purely 1D model were used, convolved with the TAS resolution function and the simulation of the two-magnon DOS is shown in Referee Fig. 1 in blue. We normalized the peak intensity of the two magnon feature to match the data—and we note that the calculated two magnon intensity is *substantially* lower than this. Nevertheless, the key feature we want to highlight is that the 11 meV mode we observe has an energy below that expected for two-magnon scattering. The single magnon peak for the primary domain is also plotted in yellow for reference. In summary, the broadened line shape of the two-magnon continuum, its relative intensity, and structure are not consistent with our data nor the long-lived lifetime of the 11 meV amplitude mode.

Referee Fig. 1: TAS data from main text Fig. 3 (c) are plotted in green symbols with the lowest energy single magnon peak plotted in yellow and two magnon continuum for a pure 1D model shown in blue.

The discussion of the Ginzburg-Landau theory is also not satisfying. Following Ref. 25, the authors discuss the lifetime and mass of the longitudinal mode in terms of the proximity to a quantum critical point (QCP). Yet the authors remark in their introduction that their compound is far from any known QCP, which renders the discussion hard to follow. I would like to remark that being far from a QCP does not imply a short-lived longitudinal mode (it is rather that one can show that in some models, being close to a QCP insures a well-defined amplitude mode). Indeed, the longitudinal mode could be long-lived because it cannot kinematically decay into two transverse modes (which, as far as I understand, could be the case here, since twice the gap of the transverse mode is larger than that of the longitudinal mode).

The referee makes a good point here. This portion of the discussion has now been substantially revised. We've focused the discussion of the mode's origin in terms of a two-magnon bound state, which as the referee notes is reflected in the fact that the amplitude mode is less than twice the energy of the zone center transverse modes. Absent theory predicting an elementary amplitude excitation in this material, this compound excitation is now the framework through which we discuss our data.

Therefore, I think that the term "anomalous" to describe the longitudinal mode should be substantiated much more. As it is, the article is too fact-focused, and I cannot recommend it for publication, unless the authors provide an improved analysis of their data.

We have tried to make this aspect clearer in the revised text. The anomalous nature of the mode comes from the fact that, to the best of our knowledge, there was no prior expectation of such a mode in a planar Heisenberg antiferromagnet with $J \gg D$ and spins that are often thought of as rather conventional. (As another referee notes, it may be inappropriate to think of $S=2$ spin as nearly classical objects, so we have revised our discussion of this somewhat) We do note that the energy scales are inconsistent with a Haldane singlet, yet the bound state seemingly arises from the quasi one dimensional nature of the spin dynamics where the magnon density of states is large. A conventional spin system that is more chemically one dimensional (well separated spin chains) such as TMMC does not show a similar mode, so the difference arises from the uniaxial anisotropy in NaMnO₂. We have now implemented a more detailed analysis of our data as well as collected new data to better parameterize the spin system.

1) I do not understand why the longitudinal and transverse energy scans are so different above T_N (Fig. 4c). Since $D \ll T_N$, and since there isn't any magnetic order, I would naively expect the fluctuations to be much more isotropic. Could the authors comment on that?

The critical scattering above T_N should naively have a divergent longitudinal component that drives the phase transition. We believe this is the origin of the anisotropy we observe at 50 K. This may seem rather far from the transition temperature for a conventional 45 K magnetic transition; however the system is dimensionally frustrated with an inherent nearest neighbor J greater than T_N . It is not uncommon to resolve an extended regime of critical scattering in frustrated systems. We have added some additional discussion to the text explaining this facet of the data.

2) Units for wave-vectors should be given.

Thank you for noting this. We have specified this in the text now.

3) The vector P in Fig. 1 is defined only later in the article.

We have now defined this Fig. 1's caption.

Reviewer #2 (Remarks to the Author):

This manuscript presents a study of the excitations in α -Na_{0.9}MnO₂. The study is motivated by two reasons: the recent success on the growth of a single crystal of this compound and two recent publications/preprint reporting so-called amplitude modes in two-dimensional (2D) systems [Tao arXiv:1705.06172v1, Jain, A. Nature Physics 13, 633–637 (2017)]. The nature of the excitations in this compound is interesting in the field of condensed matter physics, as it can give important information to further understand the phase diagram of a one-dimensional (1D) spin chain with significant single

ion anisotropy. The authors claim that the nature of these excitations can have a bigger impact on broader areas, as the findings reported by H. Tao and A. Jain do.

The authors present in this manuscript experimental results of the excitations of this compound, based on polarized neutron scattering measurements. Two gapped excitations are measured, both of them with a 1D nature. The lowest energy mode is a transverse mode, which can be understood and described using linear spin wave theory. The study of this excitation –in a powder sample- was already published by Stock, et al PRL 103, 077202 (2009). The higher energy excitation is a longitudinal mode –amplitude mode-, which has not been reported before, and its nature is unknown. The authors do not present any clear explanation on the existence of this mode and the different scenarios are shortly listed (in page 14) although they have not been explored. Which brings me to think that the authors do not understand the physics behind this compound.

I find a couple of irregularities in this manuscript and I consider that the data analysis has not been done meticulously and cannot be used to claim any conclusion, apart from just reporting the existence of the longitudinal mode. Therefore, I consider that this manuscript, in its current form, should not be published in Nature communications. My specific comments are as follow:

We thank the referee for his/her careful review of our manuscript. We have now substantially revised the paper with additional data and analysis to convey a more definitive analysis of the amplitude mode of NaMnO₂. Below, we address the referee's specific comments.

- The title is misleading as it mentions a 2D triangular antiferromagnet, but the material is an effective spin chain (1D). This is explained in the abstract where the authors claim that a new route to realize an amplitude mode in a 2D spin arrangement is to have an effective 1D magnet. This statement falls far from any logic. Please be aware that when talking about dimensionality in magnetism, we refer to the effective magnetic exchange interactions and those do not necessarily coincide with the atomic distribution in the lattice.

We have now changed the title to a “planar triangular antiferromagnet” to avoid any confusion here.

- The abstract does not represent the results. It mentions “Higgs bosons” without them being relevant in this compound as it is far away from a quantum critical point. It discusses 2D magnets, although this system is quasi-1D. It does not describe all their observations and does not provide a conclusion.

We have revised the abstract to more clearly convey our results.

- The sample reported here is not stoichiometric. However, previous published studies were done in stoichiometric powder samples. It is confusing the mixed of sample names throughout the manuscript as the stoichiometry is only discussed at the end of the manuscript. The authors should make a comment about this much earlier in the text.

We have added the stoichiometry (ie. the specific chemical formula of our sample) in the abstract to make things clearer. We do state the stoichiometry in the introduction (third paragraph) when we present our results. It however is difficult to discuss the relevance of the stoichiometry to our results until later in the text once we establish what we have observed. We have now tried to make sure the reader understands that previous results were obtained on stoichiometric powders.

- **Diffuse magnetic scattering is mentioned in page 7. The existence of this signal was also previously mentioned by Stock et al. However, it is not possible, or it is not clear in the figures where this intensity is.**

This just means that the excitations are not well defined along a given H or L (the interchain and approximate interplane directions). This now shown in the new manuscript Fig. 2 (b) and in the supplemental data.

- **The dark blue color in colorplots (Fig. 2a,c) is misleading. Please make this white as I guess there is no available data in those regions.**

This is a good idea. We have now done this.

- **Fig 2 d: The 6meV mode has an unusual sudden drop in the intensity which is not symmetric along Q. The authors make no comment regarding this.**

This was due to the drop off in small angle background of the instrument. This plot has now been moved to the supplemental section and a note regarding this drop-off has been added.

- **Have the authors looked for higher energy modes or there is only data below 13meV at (0.5,0.5,0)?**

No higher energy modes have been observed. Our new higher energy TOF data in the new Fig. 2 can be fully accounted for by the known twins and the small added J_2 value.

- **The authors compare the data to two structure factors in Fig. 3. They make their first conclusion on the nature of the higher energy mode, saying that the longitudinal function describes better the data, that making it different to a transversely polarized spin wave excitation. However, both structure factors fail completely to describe the data. There is also a clear modulation of the intensity as a function of Q –damped sine- but the authors fail on recognizing it.**

This comparison was originally meant to be somewhat qualitative and it was overly simplified. Now that we have determined the underlying J_2 of the system, the twins in the compound (which we originally neglected due to an unresolved J_2) make our original analysis of this data obsolete. Given that the polarized neutron data in Fig. 4 directly demonstrate a longitudinal polarization and the need to make space to display the new TOF data, this panel was redundant and has been removed.

- **The authors use the model –linear spin wave theory- already published by Stock in order to describe**

the dispersion relation of the transverse mode. They shortly mention in page 8 that the higher energy mode was also fit with the same J, but do not give details on the dispersion relation for this mode. Do they also use linear spin wave theory to describe this mode? They give no comment on which model they use, on how it works and how can they use a model without understanding the physics of the compound.

Our original intent was to empirically parameterize the mode's dispersion absent a microscopic model. We have removed this full fit in the revised text as we now believe that the mode's dispersion in this higher energy regime (where it merges with the transverse modes) cannot be reliably decoupled from the transverse modes of the twins. For the region near the zone center where the mode is distinctly resolvable, we fit its dispersion to the same J-D form of dispersion simply with a modified gap value. This has been made clearer in the revised text.

• In page 15 they quickly mentioned that the material is thought to be far from the Haldane regime. However, looking at DMRG results which calculate the phase diagram for s=2 Haldane chains with single ion anisotropy, the reported values of $D=0.196\text{meV}$ and $J=5.8\text{meV}$ do not look too far from the Haldane regime (Phys. Rev. B 87, 235106, 2013)

Within the reference cited by the referee, NaMnO₂ definitely lies outside of the Haldane regime, consistent with other models of S=2 Haldane systems. With J scaled to 1 meV, then the critical D value needed to enter the AF state is -0.0046 in the Heisenberg limit. Our measured D value scaled to the same notation gives a value of -0.035, nearly eight times larger than the critical threshold. Additionally, the gap is also far too large for any calculation of a S=2 Haldane state that we are aware of. We have added this reference and discussion to the text.

Reviewer #3 (Remarks to the Author):

The authors present detailed unpolarised and polarised inelastic neutron scattering results for the S = 2 triangular-lattice compound $\alpha\text{-NaMnO}_2$. They find a gapped spin-wave (transverse) mode and an additional excitation with predominantly longitudinal polarisation and a gap approximately double that of the transverse branch. They discuss the context of the amplitude (longitudinal) mode in quantum and "not-so-quantum" antiferromagnets.

This referee has no grounds for complaint concerning the experimental analysis, which appears to be solid and delivers a convincing demonstration of the existence of a longitudinal (amplitude) mode. However, the theoretical discussion is unclear and in places self-contradictory.

We thank the referee for his/her thorough review of our manuscript and for the highly constructive comments for improvement in the theoretical discussion of our data. The paper has now been substantially revised and a more coherent theoretical picture put forward. Below we address of the referee's initial question concerning the original draft.

From as early as the title, the authors refer to their material as a classical spin system. Aside from the fact that all spins are at minimum semiclassical (their origin in electron spin and their description by ladder (raising and lowering) operators), this is definitely not the case for $S = 2$. Here one may consider the clear presence of a Haldane gap in the $S = 2$ AF chain, to which the manuscript refers. If the authors wish to argue for the effects of the D term in reducing quantum fluctuations, this is certainly not clear in the early part of the discussion. Later the authors even report “substantial fluctuation effects” in the ordered moment, which reduce it significantly from its classical value.

In the revised discussion, we have clarified things. Our intent was simply to convey that $S=2$ spin systems in real materials are viewed as rather conventional---diminished local moment fluctuations and driven away from the Haldane state via anisotropies inherent to real systems.

On language, although the authors call the amplitude mode in magnetic materials a “condensed matter analogue” of the Higgs boson, this is only half an analogy because the amplitude mode lacks any gauge character (in this sense an amplitude mode of a superconducting order parameter is closer). Further, the primary new aspect of the present work is to extend the study of amplitude modes beyond the highly spin-symmetric cases discussed to date, but such an easy-axis system would seem to have only discrete symmetries, and thus the idea of a mode emerging from the breaking of a continuous symmetry is also lost.

The referee is correct in noting the lack of gauge character as a distinction from amplitude modes in spin systems from a proper Higgs mode. We have now removed the reference to the Higgs mode in the abstract as this is just a distraction from our main finding of an amplitude mode.

In several places the authors refer to their system as being “far from a quantum critical point,” also referring to a “quantum disordered regime.” First, these two concepts have to be an oxymoron in a classical spin system (first point). It is correct that an ordering transition is sufficient for the appearance of an amplitude mode, and thus they are possible in a classical system (thermal transition). However, if the authors wish to be guided by previous work on the amplitude mode near a QCP, such as Ref. [25], one of the things they will need is new language. Second, it is not at all clear that this system need be far from a 1D to 2D transition: because it has only discrete and not continuous spin symmetries, the nature of the putative QCP need not be similar to the situation in Ref. [25] and it is not a given that the mode masses would vanish at the transition. Thus the gap alone cannot be used as an indicator of proximity to a QCP.

We have revised the text to avoid these types of seeming contradictions and clarify our meaning. In the revised text we have focused our discussion in terms of a two magnon bound state driven via the uniaxial anisotropy in this material. This largely avoids the need to quantify the proximity to a QCP.

The manuscript refers both at the beginning and end to topological excitations and (bound) solitons. What do the authors have in mind here and do they really have anything to say about either topic? At minimum, some remarks of justification are required. If the former refers to Haldane-type string

order, how does this fit in with the spin being “classical” ? On the latter, again if the authors are motivated by the discussion of domain walls (also kinks or solitons) in Ising systems, several steps of logic are missing from the manuscript.

Thank you for noting this. This was meant to refer to solitons in Ising chain systems. We have clarified this in the text.

On P4 there are some curious formulations: 1) it seems to be stated that the amplitude mode is detected “as ordered state is approached,” but in fact it is only present when a magnetic order parameter is established. 2) the authors refer to a “longitudinal bound state,” but this seems to presuppose one of the possible scenarios discussed later. Is it clear, particularly in the present low-symmetry system, that the longitudinal mode has to be considered a bound mode rather than as something “elementary” ? 3) there is the first occurrence of the statement about being “far from any known QCP,” which as noted above contains two unjustified assumptions. 4) there are two unqualified statements about this being a “classical spin system.”

These points have now been corrected and clarified.

On smaller matters:

As a matter of curiosity, what is the bandwidth of the modes shown in Fig. 2a ? What happens in the upper half of the Brillouin zone ? The question of mixing with two-boson excitations is an important and potentially informative one.

The bandwidth is now shown in our new TOF data plotted as the new Figure 2 in the main text. Other than the bound state measured about 11 meV, other two-boson excitations (such as a continuum) are not resolvable.

The N'eel temperature of 45 K suggests an effective interchain interaction (in 3D) on the order of 4 meV. Can the authors explain why nothing like this is visible in Fig. 2b ? Zero effective interchain interaction due to perfect frustration should also be reflected in T_N .

Our new data has allowed us to form a J1-J2-D model describing the spin dynamics of the system. J2 is still somewhat small though (~ 0.77 meV) relative to T_N ; however the overall scale of the anisotropy is still on the order of T_n (6 meV gap).

The non-stoichiometry of the system would appear to present a significant source (10%) of disorder. The discussion of this point seems a little backwards – given that Na is a very generous electron donor, it is curious to state that its absence creates hole, and in fact thinking in a language of holes (mobile objects in semiconductors) is curious in an insulating magnet. The possible impact of 10% disorder is dismissed very quickly and this point would benefit from a deeper discussion; this referee does agree that there is no argument for the disorder to create a dispersive spin mode, and in fact

that the authors could be using it to strengthen their arguments for semiclassical behaviour [a) the system behaves as though there is little difference between the $S = 2$ and $S = 3/2$ sites; b) if the 10% reduction is included then the expected ordered moment is already rather less than $4\mu_B$ (P13), meaning that the fluctuation effect is not as strong as it appears]. However, there remains a serious non-sequitur where the authors refer to the regime of no magnetic order as “quantum disordered,” having claimed that the spin is classical.

We have tried to make this clearer in the revised manuscript. A Na vacancy creates a Mn^{4+} site simply via charge balance, which naively introduces a hole into the MnO_6 plane. The compound remains an insulator due to the enormous trapping energy of the polarons in the system. The JT distortion is quite large and Mn^{4+} locally relaxes this, creating a trapping potential. First principles calculations of the binding energy of this hole polaron state estimate it to be ~ 1 eV. So to a good approximation, it is best to think of the Na vacancies as introducing magnetic disorder into the system. We have substantially revised the discussion here mindful of the inconsistencies the referee has noted.

On P14 the authors suggest that the amplitude mode could be lighter than the gapped transverse modes. Do they have a mechanism in mind for this? If the effective binding energy of 2 transverse modes were so strong that the amplitude mode is lighter than them, should one not review theoretical picture of which excitations qualify as “elementary” and which as “compound”?

This is a good point raised by the referee. We have now focused our discussion on the amplitude mode in terms of a compound excitation rather than an elementary mode.

In the very last sentence the authors also speculate about $NaMO_2$, with no references. Do they have some solid justification in mind for this? How much is known about spin anisotropies and materials non-stoichiometries in any other such system? It seems that any change of the $S = 2$ spin will create something significantly more or less quantum than the current compound, with effects that cannot be foreseen too clearly from one known case.

We have now clarified our suggestion, which was based on the lattice geometry and coherent Jahn-Teller renormalization of the lattice into weakly coupled 1D chains. Other systems such as $NaNiO_2$, $NaFeO_2$, ect... may exhibit similar behaviors although the nature of the spin anisotropies in those materials merit further study.

In summary, the manuscript reports solid experimental work but the discussion of its context could use considerable improvement before it could be deemed suitable for publication in something like its current form.

We sincerely thank the referee for his/her support in improving the manuscript and for the many suggestions for improving the clarity of manuscript's theoretical discussion. We hope the revised

manuscript with the incorporation of additional data and theoretical modeling addresses the referee's initial concerns.

Reviewer #4 (Remarks to the Author):

The authors present evidence of a 1D longitudinally polarized magnon bound state on single crystal of the 2D antiferromagnetic material NaMnO₂. Motivated by the report of 1D spin dynamics inferred by measurement on powder samples (ref 11), the authors employ inelastic neutron scattering experiments on single crystal to unveil a dispersive mode along the b^* direction at nearly twice the energy of the lowest energy branch.

Further measurements using polarized neutron beams using different polarization direction with respect to momentum direction at the (0,3/2,0) reciprocal space position demonstrates that this mode is longitudinally polarized and has a long lifetime, limited by the instrumental resolution. This result is new, and the use of polarization analysis provides a direct experimental evidence for the polarization direction of the mode, and strongly supports the claim.

The paper is well written, concise, and certainly of interest for the magnetism community. I feel at this stage however that the publication would immensely benefit from some theoretical backup that could reproduce at least qualitatively the energy and dispersion of the mode in the (Q,w) space probed by the experiment and explain how this unusual mode differs from what founds in other antiferromagnets.

We thank the referee for his/her supportive comments and for noting the interest of our work. As detailed in our replies to reviewers 1-3, the revised manuscript now contains new data further parameterizing the spin Hamiltonian of NaMnO₂, new theoretical modeling of the origin of the amplitude mode in this material, as well as substantially revised theoretical discussion.

Alternatively, the authors need to explain why it is not possible at present to compute the dispersion and intensity of this mode with available methods or what are the current theoretical limitations. For example in the $S=1/2$ quantum system CuSO₄·5D₂O a very precise computation of the high-order spinon states had been provided and agrees remarkably well with the data (Nature Physics 9, 435–441 (2013)). Without this additional information, I am not in a position to support publication in Nature Communication at this stage.

We hope the reviewer agrees that our revised manuscript provides the needed theoretical backup advancing an understanding of the amplitude mode in NaMnO₂. There remain some details still difficult to resolve and calculate such as the precise dispersion relation for this mode; however we feel the discovery of this state and model of the two-magnon bound state in this material are of sufficient importance to merit publication in Nature Communications. We thank the referee for his/her consideration of the revised manuscript.

Reviewers' comments:

Reviewer #1 (Remarks to the Author):

I am satisfied with the new version of the manuscript. The theoretical analysis is now much more convincing. The response to my report are also satisfying, and I therefore recommend the manuscript for publication.

Reviewer #2 (Remarks to the Author):

Dear Editors and Authors

The revised Version of this manuscript is consistent. Thanks to the comments of all referees there are no dead ends, making this a solid piece of work. The Authors have given satisfactory answers to all questions and have improved the manuscript. I recommend the publication of this manuscript in Nature Communications.

Reviewer #3 (Remarks to the Author):

The authors, with their new theoretical colleague, have given the manuscript a significant overhaul, in the process answering all but one of the previous questions and raising one new one.

First on the new question, the theoretical treatment in the SM appears solid, but its presentation is somewhat cavalier in that many terms are neglected with little justification. That the final result is an effective chain model with a single-ion anisotropy is fair enough. However, then it appears that the physics of the material is that of bound-state formation in a 1D system: it has nothing to do with any (remaining, effective) interchain interaction – J_2 is thrown out quite explicitly in the first line of the treatment – and is not connected with the N'eel order. There may be a sense in which the elementary excitations can be classified as transverse and their bound state as longitudinal, but there is no order parameter in the model and therefore the bound state cannot be called an amplitude mode. Quite generally, the notion of an amplitude mode in a 1D system, which in conventional wisdom "cannot host ordered states due to quantum fluctuations" (the authors would need to make a strong point about the D term breaking all continuous symmetries), seems to be an oxymoron; the cited Refs. [6,7,27,28] are all in 2D or 3D.

This is connected with the unanswered question. The authors' answer concerning the value of T_N is entirely unconvincing. The in-chain energy scale they invoke should have very little role in the ordering temperature, which for a weakly coupled system should be related to the interchain or - plane interaction. Thus the 1D model on its own, justified by the flatness of the observed interchain band dispersions, is not consistent with the observation of magnetic order. In Ref. [11] one may read that the onset of order at 45 K is accompanied by a structural transition: according to this study, there IS a mismatch between the J_2 values for directions e_2 and e_3 in Fig. S1, perfect frustration is lifted and the model becomes 2D. It would be trivial to include this in the linear spin-wave treatment of Sec. S1 and the authors ought to be able to show that, using J_2 for direction e_2 and J_3 for e_3 , a small difference $J_2 - J_3$ is enough to make a case for order at 45 K (in principle one would also need to know $J_{\text{interplane}}$) without having a noticeable effect on the band dispersions due to the dominance of in-chain terms.

Then they have 2 possible courses of action: i) argue that the 1D model is good, by separation of energy scales (the interchain coupling scale is still well below the 1D gap), but remove all

statements related to amplitude modes, because the observed state is bound due to D and not to the magnetic order, or ii) justify the statements related to amplitude modes by keeping $J_2 - J_3$ in the effective model and showing that it does have some role in promoting the binding. In case (i), the loss of the bound-state mode intensity at 50 K [Fig. 4(c)] is a coincidence, which might even be expected on energy-scale grounds; to justify case (ii), some measurements at additional temperature points would be required (and would in any case be helpful for the physical understanding).

On a minor matter of terminology, "easy-axis" has the implication of an easy-axis bond interaction, i.e. an Ising-type $S_i S_j$ (Hamiltonian) term, rather than a single-ion anisotropy ($D S_j^2$). Here the authors seem to mean only the latter but use both (linguistic) terms without distinguishing very carefully, which is confusing in the early part of the manuscript.

On an apparently minor matter of consistency, if one reads the expressions for the magnon energy obtained from spin-wave theory, one would expect the upper band edge to be close to $2J_1$, whereas from Fig. 2 it is manifestly $4J_1$. Perhaps the authors have forgotten some factors of S in the formulae they present.

In summary, the manuscript presents two good stories, namely 1D bound states and order-parameter amplitude modes, but for the moment there is a fundamental mismatch which will need to be fixed before it could be acceptable for publication.

Dear Editor,

Thank you very much for forwarding us the latest round of reviewer comments on our manuscript. We have now addressed the final comments from referee #3 in the revised text as well as provided detailed replies to the questions raised by the same referee below. We feel that the revised text is now suitable for publication in Nature Communications. Thank you very much for your consideration.

Reviewers' comments:

Reviewer #1 (Remarks to the Author):

I am satisfied with the new version of the manuscript. The theoretical analysis is now much more convincing. The response to my report are also satisfying, and I therefore recommend the manuscript for publication.

We thank the reviewer for his/her continued review of the manuscript and recommendation for publication in Nature Communications.

Reviewer #2 (Remarks to the Author):

The revised Version of this manuscript is consistent. Thanks to the comments of all referees there are no dead ends, making this a solid piece of work. The Authors have given satisfactory answers to all questions and have improved the manuscript. I recommend the publication of this manuscript in Nature Communications.

We thank the referee for his/her recommendation for publication.

Reviewer #3 (Remarks to the Author):

The authors, with their new theoretical colleague, have given the manuscript a significant overhaul, in the process answering all but one of the previous questions and raising one new one.

We thank the referee for his/her continued review of the manuscript answer the remaining questions in the paragraphs below.

First on the new question, the theoretical treatment in the SM appears solid, but its presentation is somewhat cavalier in that many terms are neglected with little justification.

The calculation we present indeed neglects terms that are small provided $Je^{-S} \ll D \ll J$ and $J_2 \ll J$. This is a reasonable approximation given the experimentally determined values and stated in the

supplemental information section. We have now added further text to the main manuscript ensuring that this is made clear.

That the final result is an effective chain model with a single-ion anisotropy is fair enough. However, then it appears that the physics of the material is that of bound-state formation in a 1D system: it has nothing to do with any (remaining, effective) interchain interaction – J_2 is thrown out quite explicitly in the first line of the treatment – and is not connected with the Néel order. There may be a sense in which the elementary excitations can be classified as transverse and their bound state as longitudinal, but there is no order parameter in the model and therefore the bound state cannot be called an amplitude mode. Quite generally, the notion of an amplitude mode in a 1D system, which in conventional wisdom “cannot host ordered states due to quantum fluctuations” (the authors would need to make a strong point about the D term breaking all continuous symmetries), seems to be an oxymoron; the cited Refs. [6,7,27,28] are all in 2D or 3D.

The referee’s first statement is true: it is the physics of a 1d system, and there will be only small corrections due to J_2 . The second statement, however, that the model results are not connected to Néel order is incorrect. Our theoretical approach is that of a spin wave expansion about an ordered state. The system is ordered, even though 1d, because the calculation is at zero temperature and there is an Ising anisotropy. This is enough to stabilize order in a purely 1d system at zero temperature and preserves the notion of an amplitude mode.

This is connected with the unanswered question. The authors’ answer concerning the value of T_N is entirely unconvincing. The in-chain energy scale they invoke should have very little role in the ordering temperature, which for a weakly coupled system should be related to the interchain or - plane interaction. Thus the 1D model on its own, justified by the flatness of the observed interchain band dispersions, is not consistent with the observation of magnetic order.

We can discuss T_N for this situation. In the purely 1D system there is order at $T=0$, and at $T>0$ there is a large correlation length which forms approximately as $\xi \sim e^{\frac{\sqrt{DJ}}{k_B T}}$. Given this estimate, the inter-chain energy due to J_2 is of order $J_2 \xi$ and order occurs when $J_2 \xi > k_B T$. This can be rewritten as $\xi > \frac{k_B T}{J_2}$ and taking the log of this we have $\frac{\sqrt{DJ}}{k_B T} > \ln\left(\frac{k_B T}{J_2}\right)$ or $k_B T < \sqrt{DJ} / \ln\left(\frac{k_B T}{J_2}\right)$ which approximately gives

$$k_B T_N \sim \sqrt{DJ} / \ln\left(\sqrt{\frac{DJ}{J_2}}\right)$$

While this rough estimate lacks the exact prefactor, one can see that although indeed $T_N \rightarrow 0$ as $J_2 \rightarrow 0$, this occurs only via a very slow logarithmic factor. So T_N may be quite large even for small J_2 . However, the mode dispersion is not much affected by J_2 . While the referee is correct that J_2 cannot be neglected in determining T_N , this is a very weak effect, and the neglect is a good approximation for understanding the modes in the system.

In Ref. [11] one may read that the onset of order at 45 K is accompanied by a structural transition: according to this study, there IS a mismatch between the J_2 values for directions e_2 and e_3 in Fig.

S1, perfect frustration is lifted and the model becomes 2D. It would be trivial to include this in the linear spin-wave treatment of Sec. S1 and the authors ought to be able to show that, using J_2 for direction e_2 and J_3 for e_3 , a small difference $J_2 - J_3$ is enough to make a case for order at 45 K (in principle one would also need to know $J_{\text{interplane}}$) without having a noticeable effect on the band dispersions due to the dominance of in-chain terms.

We looked very carefully in order to attempt and resolve the previously reported triclinic distortion in our crystal and within resolution there is no structural distortion (symmetry lowering) down to 2K. At a minimum such a distortion is absent in our $\text{Na}_{0.9}\text{MnO}_2$ crystal, perhaps suppressed by the Na vacancies in the sample.

Then they have 2 possible courses of action: i) argue that the 1D model is good, by separation of energy scales (the interchain coupling scale is still well below the 1D gap), but remove all statements related to amplitude modes, because the observed state is bound due to D and not to the magnetic order, or ii) justify the statements related to amplitude modes by keeping $J_2 - J_3$ in the effective model and showing that it does have some role in promoting the binding. In case (i), the loss of the bound-state mode intensity at 50 K [Fig. 4(c)] is a coincidence, which might even be expected on energy-scale grounds; to justify case (ii), some measurements at additional temperature points would be required (and would in any case be helpful for the physical understanding).

As discussed above, the identification of an amplitude mode in our model is still correct. We do not believe the additional terms are essential/helpful to include, nor do they effect the result. Not to lose sight of the key result, at its core our paper is the experimental identification of a longitudinally polarized, dispersive mode in the AF ordered state of a planar AF where frustration renormalizes the spin physics to be quasi one dimensional. The mode is empirically the oscillation of the amplitude of the ordered moment, and our supporting model is the minimal picture that captures the emergence of the bound state we observe. While of interest in future studies, tracking the detailed temperature dependence of the mode is nontrivial due to the critical fluctuations that arise as T_N is neared and is beyond the scope of the present paper.

On a minor matter of terminology, “easy-axis” has the implication of an easy-axis bond interaction, i.e. an Ising-type $S_i S_j$ (Hamiltonian) term, rather than a single-ion anisotropy (DS_i^2). Here the authors seem to mean only the latter but use both (linguistic) terms without distinguishing very carefully, which is confusing in the early part of the manuscript.

We mean “easy-axis” as a reference to the tendency of the system to order spins along a particular axis, rather than perpendicular to it. This is indeed what the single ion term does. To avoid confusion, we have sharpened this to now state "uniaxial single ion anisotropy".

On an apparently minor matter of consistency, if one reads the expressions for the magnon energy obtained from spin-wave theory, one would expect the upper band edge to be close to $2J_1$, whereas from Fig. 2 it is manifestly $4J_1$. Perhaps the authors have forgotten some factors of S in the formulae

they present.

Thank you for noting this. There was a typo and a factor of S missing in the main text's formula.

In summary, the manuscript presents two good stories, namely 1D bound states and order-parameter amplitude modes, but for the moment there is a fundamental mismatch which will need to be fixed before it could be acceptable for publication.

As we hope we have detailed above, there is not a direct contradiction between the notion of a 1D bound state and the amplitude mode in our model. We hope the additional clarifications made to the manuscript text and our answers to the referee's questions now convince the referee that our manuscript is suitable for publication in Nature Communications.

Reviewers' comments:

Reviewer #1 (Remarks to the Author):

Dear Editor,

Having read both the referees' reports and the authors' response, I tend to agree with the authors, although I would recommend that they make some adjustment to their manuscript.

In their second report, Referee #3 has two main concerns:

1) Is the system effectively 1D (as discussed in the text and supplementary information), but then there cannot be any amplitude fluctuation of the order parameter (even with the single ion anisotropy, at finite temperature), which is in contradiction with the idea of an amplitude mode; Or is the system really 2D (which would allow order), but then the text and supplementary information are completely misleading.

2) Is the physics instead related to the structural transition seen in Ref. 11, which would render the system 2D (thus allowing for an order parameter and its amplitude fluctuations). This however should be discussed in the theoretical treatment.

My summary of my understanding of the authors' reply, and the information found in the manuscript and the SN is:

1) The system is quasi-1D, meaning that the coupling along the chains is much stronger than the inter-chain coupling (which is much stronger than the inter-plane coupling, completely neglected here). Furthermore, the single ion anisotropy D renders the system at low energy to be Ising-like. Since at zero-temperature a quantum Ising model is ordered, one can neglect the inter-chain interaction. The authors thus perform a large S expansion around the ordered state (which is along the z -axis), which, along a ladder resummation, allows them to give a theoretical estimate of the bound-state mass.

At finite temperature, the issue is subtler, since in principle thermal fluctuations should destroy this ordered state. However, as pointed out by the authors, the quasi-1D-ness of the system should then be invoked, as J_2 is then sufficient to stabilize the order, possibly up to a relatively high temperature. If one is only interested in a temperature range much smaller than the critical temperature, it is nevertheless quite natural to just assume an ordered state and neglect temperature effects (since temperature will not qualitatively change the physics). One expects that the main effect of finite temperature will appear as a dependence of the various physical quantities on the ratio J_2/T (for systems with exponentially large correlation length at finite temperature, one usually expects a rather weak dependence).

If this is indeed an accurate presentation of the authors' line of thoughts, it should be made clearer, and discussed, at least in the supplementary information (indeed, all the finite temperature effects were only discussed in the reply to Referee #3's report, and the other aspects of this reasoning are not clearly explained in the text or in the supplementary information).

2) Concerning the Referee's second point, the authors just state that they do not observe this structural transition, contrary to what is reported in Ref. 11. This does not appear to be discussed in the text.

In their third report, Referee #3 seems to have convinced themselves that an AF order is possible in the 1D model used by the authors. The Referee then questions whether the values of J and D obtained experimentally insure that the model does not have a Haldane state.

On this point, Ref 13 says that the Haldane phase is destroyed for $D/J=0.04(2)$, which is pretty

much the experimental value $(D/J)_{\text{exp}}=0.035$ (it is not clear why the authors write that the compound is thought not to have a Haldane state. Do they have any additional references?). However, the Haldane gap quoted in Ref. 13 is about $0.08J$, much smaller than the one observed. Finally, the model Eq. (15) of the Supplementary Information is an effective one (after neglecting J_2), and the fate of the Haldane state in that case is not obvious to me, although it is of course an important question.

The Referee also complains that the AF order is not discussed in Sec. S2. I disagree with the Referee. Sec. S2 clearly follows on from Sec. S1. In Sec. S1, the authors use the J_1 - J_2 -D model to obtain the dispersion relation of excitations around the mean-field ordered AF state. In Sec. S2, following the same reasoning than in Sec. S1, but neglecting J_2 , they compute the longitudinal mode mass. Thus, they indeed start from an ordered AF state (followed by a large S + RPA approximation).

The Referee then addresses the point 2 of their first report, and complains that the authors have not addressed it properly. As I have stated above, it is indeed true that the reply by the authors is rather brief (and not addressed in the text). This point deserves to be discussed in a way or another in the manuscript, although I am ready to trust the authors when they say they do not see this transition in their compound. I do not feel that this is a good enough reason to deny publication.

In summary, while the authors' reply to Referee #3 second report are satisfying to me, the change made to the manuscript were too minimal in my opinion. In particular, I find that the main issue of the Referee (point 1) has been addressed properly by the authors in their reply. However, the authors should seriously consider writing a clearer discussion of their reasoning in applying the different approximations (both at zero and finite temperature). This should be understandable to any reader, even if they are not able to follow the calculations themselves.

Concerning point 2, I agree with the Referee that this should also be discussed somewhere. However, I do not see this point as a major issue that should prevent publication.

As a minor comment, I agree with the Referee that the term "Ising-like" is not clear. The authors could just state (in a way or another) that D induces order along a privileged axis (z-axis), rendering the system effectively described by an Ising order parameter / model at low energy.

Reviewer #2 (Remarks to the Author):

Comments on previous reviews from the experimental point of view:

- Longitudinal mode existence: The manuscript shows clear experimental evidence of the existence of a longitudinal mode in $\text{Na}_0.9\text{MnO}_2$

- Dimensionality: Previous inelastic neutron scattering (INS) study in powder samples [Phys. Rev. Lett. 103, 077202] (from the same authors of the neutron powder diffraction paper) show that the only way to account for the INS spectral weight is to consider 1D magnetic correlations instead of the 2D correlations seen before in the powder diffraction [PRL 99, 247211 (2007)]. INS is indeed a much more powerful tool to tell the dimensionality of magnetic correlations than diffraction. The flat modes along the L-axis are indeed a sign of null or extremely weak magnetic exchange interactions along those directions. Due to the overall nature of the excitations, the system can be considered as quasi-1D magnet, in which interchain interactions become important below the Neel temperature.

- Crystal structure: The authors claim not to find evidence of a further distortion from Monoclinic to Triclinic. This distortion appears to be very small [PRL 99, 247211 (2007)] and it could be that it is only possible to detect by using high resolution diffraction techniques. However this distortion is not fundamental in the current discussion and it could be studied and discussed in further

publications. The authors should mention this in their manuscript and not only mention the Jahn Teller distortion.

- Further experiments: This works opens up the possibility for further studies to answer the following questions: What is the temperature dependence of the longitudinal mode? What is its behavior in a transverse field? Are there further bound states at higher energies? What is the influence of Na content in the crystal structure of the compound? Some of these questions have been asked by Referee 3. However, I consider these, although interesting, do not need to be answer in the current manuscript.

- Experimental challenges: The authors have known to overcome experimental difficulties. The sample is twinned, the magnetic signal is weak, the magnetic signal extends over long energy scales (relative to INS) making difficult to find the best settings in terms of energy resolution, wave vector resolution and intensity. However, the authors were able to do a fairly good analysis of the data.

Comments on previous reviews from the proposed model point of view:

- The existence of the longitudinal mode in the ordered phase cannot be explained by Spin Wave theory. Further models are necessary to explain their presence. DMRG methods have been shown to fail due to the growth of entanglement in these type of systems [Phys. Rev. B 96, 054423]. The authors present a perturbative calculation not as the real solution for the system, but more in the manner of an exercise. Where, they proof the possibility of the emergence of a bound state due to an attractive potential originated in the difference between the single ion anisotropy and the nearest neighbor magnetic interaction. This model however makes big assumptions and treats J_2 as a small perturbation, even in the Neel state. This model should not be shown as the main subject of the manuscript but just as a side exercise.

- As it is shown experimentally, the emergence of the longitudinal mode is linked to the Neel temperature and therefore related to the ordered state (if not, and as the Referee 3 says, it will be a big coincidence). The confining potential should be proportional to the magnitude of the ordered moment and therefore it should be linked to the interchain magnetic exchange interaction J_2 .

Suggestions on how to proceed:

- The authors should consider some of the observations of Referee 3 and do changes in the discussion accordingly. This is: briefly comment on the crystal structure and its relevance for the current results. Propose a second model (or add on a discussion) where J_2 is consider not as a perturbation but it is acting as a staggered magnetic field that confines the bound states.

- In the revised version they should write the sample temperature in Figures 2 and 3.

Reviewer #3 (Remarks to the Author):

The authors have made minimal changes to their presentation. They should therefore not be surprised that the referee's opinion has not changed very much. The manuscript remains a confused and confusing mixture of disconnected and inadequately explained ideas.

On the plus side, this referee now understands that the chain model of Eq. (15) of the SI can have an AF ordered phase. This understanding was brought on not by any efforts on the part of the authors to explain the situation, which were based only on assertion, but by considering the phase diagram of M. Den Nijs and K. Rommelse, Phys. Rev. B. 40, 4709 (1989) for the $S = 1$ case. Given that the $S = 2$ Heisenberg case also has a Haldane state, a finite $|D|$ (negative, i.e. easy-axis,

single-ion anisotropy) is required. Can the authors show that D is large enough in the present case ($D/J = 0.215/6.16 = 0.035$) ? The quoted references [13,25,26] seem not to answer this question. Does the model of Eq. (15) have a D_c (critical value of D for AF order) ? The entire analysis of Sec. S2 appears to remain entirely disconnected from the issue of AF order, discussing as it does only the formation of bound states. Where can one see the AF order parameter entering the discussion ? Without this it is hard to claim that the longitudinal mode is also an amplitude mode of this order parameter.

On the minus side, the authors have done nothing to address (in the manuscript or the referee reply) whether this is really physics of the system. It is not acceptable simply to ignore the previous structural analysis, given that this appears to have been performed by a more accurate experimental method. Did the present work study the lattice structure at all, and if so was it by high-resolution diffractometry ? This is not clear from anything in the manuscript or the SI. Do the authors wish to claim that an alternative explanation for the previous results exists, e.g. by an interpretation of the twinning of magnetic domains ? Or do they wish to claim that the non-stoichiometry of the present sample removes the distortion found in the previous study ? If this is the case, the authors will need to be much more explicit – in the manuscript, and not just for the referee – because glib and unqualified statements of the “we did not see this” type add nothing to the overall understanding of the system.

The definition of the language has been improved, but applying “Ising-like” to a single spin remains inaccurate and the authors will need to define what they mean by this.

Neither the reply letter nor the manuscript addressed the previous comment concerning the authors' claims about the temperature-dependence of their longitudinal signal.

However the above two issues are resolved, the discussion in the manuscript remains misleading: the transverse and longitudinal modes are properties of the chain model, as shown in detail in Sec. S2, and have nothing to do with the interchain coupling, but this statement can still be found in the text.

In summary, this referee agrees with the identification of a longitudinal mode. The manuscript further identifies a suitable 1D model and shows that this mode is a bound state of 2 magnons. It does NOT show that this mode is the amplitude mode of an AF order parameter. If the authors can extend their analysis to show this, and provide a definitive discussion of the structural question, then the manuscript would become acceptable for publication. However, in its current state it remains incomplete due to the still-unaddressed mismatch identified in the previous report.

Reviewer #1 (Remarks to the Author):

Having read both the referees' reports and the authors' response, I tend to agree with the authors, although I would recommend that they make some adjustment to their manuscript.

In their second report, Referee #3 has two main concerns:

1) Is the system effectively 1D (as discussed in the text and supplementary information), but then there cannot be any amplitude fluctuation of the order parameter (even with the single ion anisotropy, at finite temperature), which is in contradiction with the idea of an amplitude mode; Or is the system really 2D (which would allow order), but then the text and supplementary information are completely misleading.

2) Is the physics instead related to the structural transition seen in Ref. 11, which would render the system 2D (thus allowing for an order parameter and its amplitude fluctuations). This however should be discussed in the theoretical treatment.

My summary of my understanding of the authors' reply, and the information found in the manuscript and the SN is:

1) The system is quasi-1D, meaning that the coupling along the chains is much stronger than the inter-chain coupling (which is much stronger than the inter-plane coupling, completely neglected here). Furthermore, the single ion anisotropy D renders the system at low energy to be Ising-like. Since at zero-temperature a quantum Ising model is ordered, one can neglect the inter-chain interaction. The authors thus perform a large S expansion around the ordered state (which is along the z -axis), which, along a ladder resummation, allows them to give a theoretical estimate of the bound-state mass.

At finite temperature, the issue is subtler, since in principle thermal fluctuations should destroy this ordered state. However, as pointed out by the authors, the quasi-1D-ness of the system should then be invoked, as J_2 is then sufficient to stabilize the order, possibly up to a relatively high temperature. If one is only interested in a temperature range much smaller than the critical temperature, it is nevertheless quite natural to just assume an ordered state and neglect temperature effects (since temperature will not qualitatively change the physics). One expects that the main effect of finite temperature will appear as a dependence of the various physical quantities on the ratio J_2/T (for systems with exponentially large correlation length at finite temperature, one usually expects a rather weak dependence).

If this is indeed an accurate presentation of the authors' line of thoughts, it should be made clearer, and discussed, at least in the supplementary information (indeed, all the finite temperature effects were only discussed in the reply to Referee #3's report, and the other aspects of this reasoning are not clearly explained in the text or in the supplementary information).

As the referee requested, we have now added clarification in the main text explaining our neglect of J_2 in the spin chain model as well why the system can be thought to be quasi-1D at low temperatures.

2) Concerning the Referee's second point, the authors just state that they do not observe this structural transition, contrary to what is reported in Ref. 11. This does not appear to be discussed in the text.

We have now added a brief discussion of this aspect to the main text. Briefly, the reported transition in NaMnO₂ is very subtle and not entirely consistent among reports in the literature. Samples synthesized by the same group later claimed that only some small fraction of the material undergoes this triclinic distortion within a highly inhomogeneous state (Zorko et al. Nature Communications 5, 3222 (2014)). Given the focus and resolution of our studies, we can only state that we did not see the distortion within resolution; however the distortion is proposed to be very subtle. We have clarified in the text that the reported distortion is in fact very small (0.12%), not observed in our samples, and not likely to generate the anisotropy needed to justify a J_1 - J_2 - J_3 model.

In their third report, Referee #3 seems to have convinced themselves that an AF order is possible in the 1D model used by the authors. The Referee then questions whether the values of J and D obtained experimentally insure that the model does not have a Haldane state. On this point, Ref 13 says that the

Haldane phase is destroyed for $D/J=0.04(2)$, which is pretty much the experimental value (D/J)_{exp}=0.035 (it is not clear why the authors write that the compound is thought not to have a Haldane state. Do they have any additional references?).

The paper referred to by the referee here (Ref. 13) states the bound for the Haldane state as $D/J=0.04$ for the case of easy-plane (not easy-axis) anisotropy. In our case, D is reflective of easy-axis anisotropy and the calculated bound is $D/J= -0.0046$ from Ref. 26. Our D value is nearly 8 times larger than this threshold value. We discussed this point in our original reply to Referee 2, and we have now added this number explicitly to the revised main text for additional clarity.

However, the Haldane gap quoted in Ref. 13 is about $0.08J$, much smaller than the one observed. Finally, the model Eq. (15) of the Supplementary Information is an effective one (after neglecting J_2), and the fate of the Haldane state in that case is not obvious to me, although it is of course an important question.

As the referee notes, the gap value is much too large for any calculations of a $S=2$ Haldane state. Hence our results do not consider the Haldane instability as the operative physics here.

The Referee also complains that the AF order is not discussed in Sec. S2. I disagree with the Referee. Sec. S2 clearly follows on from Sec. S1. In Sec. S1, the authors use the J_1 - J_2 - D model to obtain the dispersion relation of excitations around the mean-field ordered AF state. In Sec. S2, following the same reasoning than in Sec. S1, but neglecting J_2 , they compute the longitudinal mode mass. Thus, they indeed start from an ordered AF state (followed by a large S + RPA approximation).

Thank you for noting this. Our theoretical treatment is indeed an expansion about an ordered state.

The Referee then addresses the point 2 of their first report, and complains that the authors have not addressed it properly. As I have stated above, it is indeed true that the reply by the authors is rather brief (and not addressed in the text). This point deserves to be discussed in a way or another in the manuscript, although I am ready to trust the authors when they say they do not see this transition in their compound. I do not feel that this is a good enough reason to deny publication.

We now address the lack of a resolvable structural distortion in our system within the main text. We further comment that any such distortion will be very small and unlikely to yield meaningful exchange anisotropies. The Zorko et al. Nature Communications 5, 3222 (2014) reference has been added, which addresses this point.

In summary, while the authors' reply to Referee #3 second report are satisfying to me, the change made to the manuscript were too minimal in my opinion. In particular, I find that the main issue of the Referee (point 1) has been addressed properly by the authors in their reply. However, the authors should seriously consider writing a clearer discussion of their reasoning in applying the different approximations (both at zero and finite temperature). This should be understandable to any reader, even if they are not able to follow the calculations themselves. Concerning point 2, I agree with the Referee that this should also be discussed somewhere. However, I do not see this point as a major issue that should prevent publication.

These points are now explicitly addressed in the revised main text.

As a minor comment, I agree with the Referee that the term “Ising-like” is not clear. The authors could just state (in a way or another) that D induces order along a privileged axis (z-axis), rendering the system effectively described by an Ising order parameter / model at low energy.

This point of clarification has now been added.

Reviewer #2 (Remarks to the Author):

Comments on previous reviews from the experimental point of view:

- Longitudinal mode existence: The manuscript shows clear experimental evidence of the existence of a longitudinal mode in Na_{0.9}MnO₂

- Dimensionality: Previous inelastic neutron scattering (INS) study in powder samples [Phys. Rev. Lett. 103, 077202] (from the same authors of the neutron powder diffraction paper) show that the only way to account for the INS spectral weight is to consider 1D magnetic correlations instead of the 2D correlations seen before in the powder diffraction [PRL 99, 247211 (2007)]. INS is indeed a much more powerful tool to tell the dimensionality of magnetic correlations than diffraction. The flat modes along the L-axis are indeed a sign of null or extremely weak magnetic exchange interactions along those directions. Due to the overall nature of the excitations, the system can be considered as quasi-1D magnet, in which interchain interactions become important below the Neel temperature.

Thank you for noting this.

- Crystal structure: The authors claim not to find evidence of a further distortion from Monoclinic to Triclinic. This distortion appears to be very small [PRL 99, 247211 (2007)] and it could be that it is only possible to detect by using high resolution diffraction techniques. However this distortion is not fundamental in the current discussion and it could be studied and discussed in further publications. The authors should mention this in their manuscript and not only mention the Jahn Teller distortion.

As mentioned in our reply to referee 1, discussion of this potential distortion has now been added to the manuscript.

- Further experiments: This work opens up the possibility for further studies to answer the following questions: What is the temperature dependence of the longitudinal mode? What is its behavior in a transverse field? Are there further bound states at higher energies? What is the influence of Na content in the crystal structure of the compound? Some of these questions have been asked by Referee 3. However, I consider these, although interesting, do not need to be answered in the current manuscript.

- Experimental challenges: The authors have known to overcome experimental difficulties. The sample

is twinned, the magnetic signal is weak, the magnetic signal extends over long energy scales (relative to INS) making difficult to find the best settings in terms of energy resolution, wave vector resolution and intensity. However, the authors were able to do a fairly good analysis of the data.

Thank you for noting this.

Comments on previous reviews from the proposed model point of view:

- The existence of the longitudinal mode in the ordered phase cannot be explained by Spin Wave theory. Further models are necessary to explain their presence. DMRG methods have been shown to fail due to the growth of entanglement in these type of systems [Phys. Rev. B 96, 054423]. The authors present a perturbative calculation not as the real solution for the system, but more in the manner of an exercise. Where, they proof the possibility of the emergence of a bound state due to an attractive potential originated in the difference between the single ion anisotropy and the nearest neighbor magnetic interaction. This model however makes big assumptions and treats J_2 as a small perturbation, even in the Neel state. This model should not be shown as the main subject of the manuscript but just as a side exercise.

In fact the calculation in S2 is a spin wave expansion, just carried out beyond harmonic order. We actually believe the model is not such a crude approximation. It does only direct address zero temperature, and indeed neglects the J_2 coupling. However, the J_2 coupling is indeed small, and at zero temperature J_2 is not necessary to stabilize the ordered state (the single ion anisotropy does this). Anyway, the model is already presented in the supplementary material, so is hardly the main subject of the manuscript.

- As it is shown experimentally, the emergence of the longitudinal mode is linked to the Neel temperature and therefore related to the ordered state (if not, and as the Referee 3 says, it will be a big coincidence).

We don't disagree that the longitudinal mode is linked to the ordered state. The theoretical calculation is a spin wave expansion, and so is inextricably tied to the assumption of order. It is definitely true that a $T > 0$, some inter-chain coupling is needed to stabilize the order, so if we were to do a $T > 0$ calculation we would need to include J_2 . However, this is a much more difficult calculation and we did not see that it would add much to understanding. Conceptually, the situation is fairly clear, we think: once antiferromagnetic order is established, spin waves become well-defined quasiparticles. A pair of such quasiparticles may bind if they attract, and this is a longitudinal excitation. An attraction is provided by the single-ion uniaxial anisotropy. None of these points requires pure one dimensionality.

The confining potential should be proportional to the magnitude of the ordered moment and therefore it should be linked to the interchain magnetic exchange interaction J_2 .

Please see the response below.

Suggestions on how to proceed:

- The authors should consider some of the observations of Referee 3 and do changes in the discussion accordingly. This is: briefly comment on the crystal structure and its relevance for the current results. Propose a second model (or add on a discussion) where J_2 is considered not as a perturbation but it is acting as a staggered magnetic field that confines the bound states.

We have expanded the discussion in order to hopefully clarify the physics in response to the questions about the role of J_2 and the nature of the calculation in S2.

There were a couple of suggestions concerning J_2 as a “confining potential” or a “staggered magnetic field that confines the bound states”. As far as we understand it, this refers to superficially similar situations that occur in weakly coupled $S=1/2$ Heisenberg chains, or weakly coupled transverse field Ising chains, in which interchain coupling may be treated by a “chain mean field theory” as a self-consistent staggered field on the spins of a chain, whose magnitude is proportional to the strength of the interchain coupling and the ordered moment. In those situations, the effect of such a staggered field is dramatic because it leads to *confinement*. This is because the elementary excitations of those types of chains, without the field, are *spinons*, which are fractional particles that are totally different from spin waves. These particles become confined, i.e. infinity strongly bound together, due to the staggered field, because they have a fragile and intrinsically one-dimensional nature, which is spoiled by the interchain coupling.

In our case, one could indeed include J_2 and treat it via chain mean field theory as a staggered field. However, since we are dealing with an ordered state, even without J_2 , the elementary excitations we work with are spin waves, and the longitudinal mode is a bound state of them even without the J_2 . Furthermore, spin waves are not particular to 1d, and they are not confined by the J_2 term. So there is no mechanism for J_2 to produce bound states. It is not especially relevant to the bound state formation and indeed this is why (in addition to simplicity of the calculation) we did not include it.

- In the revised version they should write the sample temperature in Figures 2 and 3.

Thanks for noting this. While these temperatures were stated in the main text, these temperatures have now also been added to the figure captions.

Reviewer #3 (Remarks to the Author):

The authors have made minimal changes to their presentation. They should therefore not be surprised that the referee's opinion has not changed very much. The manuscript remains a confused and confusing mixture of disconnected and inadequately explained ideas.

On the plus side, this referee now understands that the chain model of Eq. (15) of the SI can have an AF ordered phase. This understanding was brought on not by any efforts on the part of the authors to explain the situation, which were based only on assertion, but by considering the phase diagram of M. Den Nijs and K. Rommelse, Phys. Rev. B. 40, 4709 (1989) for the $S = 1$ case. Given that the $S = 2$ Heisenberg case also has a Haldane state, a finite $|D|$ (negative, i.e. easy-axis, single-ion anisotropy)

is required. Can the authors show that D is large enough in the present case ($D/J = 0.215/6.16 = 0.035$) ? The quoted references [13,25,26] seem not to answer this question.

We feel that this is not an accurate or fair statement by the referee. Ref. 26 explicitly shows that our measured D value is well outside the calculated range for the Haldane state (8 times larger than the lower bound). This was explicitly discussed in our original reply to referee 2. As detailed earlier, we have now added the numerical comparison to the boundaries calculated in Ref. 26 to the main text. Additionally, all quoted references (13, 25, 26) are appropriate and show that the $S=2$ 1D chain model with single ion anisotropy can have an AF ordered state.

Does the model of Eq. (15) have a D_c (critical value of D for AF order) ? The entire analysis of Sec. S2 appears to remain entirely disconnected from the issue of AF order, discussing as it does only the formation of bound states. Where can one see the AF order parameter entering the discussion ? Without this it is hard to claim that the longitudinal mode is also an amplitude mode of this order parameter.

The critical value for the 1D Heisenberg J_1 -D model has been calculated in Ref. 26. Again, we discussed this in our original reply to referee 2. Our calculations are performed about the ordered AF ground state, and we thought this progression is quite clear (as referee 2 notes). Just to be sure, we have added the statement to the opening of Supplementary Note in the supplemental material. "In this section, we expand about the 0 K antiferromagnetic ground state and ignore the coupling J_2 between chains for simplicity,.."

On the minus side, the authors have done nothing to address (in the manuscript or the referee reply) whether this is really physics of the system. It is not acceptable simply to ignore the previous structural analysis, given that this appears to have been performed by a more accurate experimental method. Did the present work study the lattice structure at all, and if so was it by high-resolution diffractometry ? This is not clear from anything in the manuscript or the SI. Do the authors wish to claim that an alternative explanation for the previous results exists, e.g. by an interpretation of the twinning of magnetic domains ? Or do they wish to claim that the non-stoichiometry of the present sample removes the distortion found in the previous study ? If this is the case, the authors will need to be much more explicit – in the manuscript, and not just for the referee – because glib and unqualified statements of the "we did not see this" type add nothing to the overall understanding of the system.

As detailed in our replies to referees 1 and 2, we have now added an explicit discussion regarding the apparent absence of the triclinic distortion in our sample.

The definition of the language has been improved, but applying "Ising-like" to a single spin remains inaccurate and the authors will need to define what they mean by this.

We have now added a statement qualifying what we mean by "Ising-like".

Neither the reply letter nor the manuscript addressed the previous comment concerning the authors' claims about the temperature-dependence of their longitudinal signal.

Our previous reply did address the referee's request for more data. We argued that this is not a reasonable request for the current manuscript and it is best left for future studies by saying "While of interest in future studies, tracking the detailed temperature dependence of the mode is nontrivial due to the critical fluctuations that arise as T_N is neared and is beyond the scope of the present paper." The other two referees are in agreement with our stance here.

However the above two issues are resolved, the discussion in the manuscript remains misleading: the transverse and longitudinal modes are properties of the chain model, as shown in detail in Sec. S2, and have nothing to do with the interchain coupling, but this statement can still be found in the text.

We presume that the referee is referring to the general statement in the text "For an easy-axis antiferromagnetic chain coupled via weak interchain coupling, two degenerate, gapped transverse magnon modes are expected as Néel order sets in;..." To avoid confusion we have changed this to "For an easy-axis antiferromagnetic chain at 0 K, two degenerate, gapped transverse magnon modes are expected as Néel order sets in;..."

In summary, this referee agrees with the identification of a longitudinal mode. The manuscript further identifies a suitable 1D model and shows that this mode is a bound state of 2 magnons. It does NOT show that this mode is the amplitude mode of an AF order parameter. If the authors can extend their analysis to show this, and provide a definitive discussion of the structural question, then the manuscript would become acceptable for publication. However, in its current state it remains incomplete due to the still-unaddressed mismatch identified in the previous report.

As discussed and agreed by the referee above, the model we use indeed has an AF ordered state and our subsequent calculations unveil the presence of a bound state of two magnons arising from the ordered state. We propose this to be a minimal model capturing the appearance of the longitudinal mode we observe. We have tried to make this clearer in the revised text and supplemental.